# Telomere-to-telomere DNA replication timing profiling using single-molecule sequencing with Nanotiming

Bertrand Theulot [1,2,5], Alan Tourancheau [1], Emma Simonin Chavignier[1], Etienne Jean[1], Jean-Michel Arbona [3,6], Benjamin Audit [4], Olivier Hyrien [1], Laurent Lacroix [1] & Benoît Le Tallec [1] ✉

Current temporal studies of DNA replication are either low-resolution or require complex cell synchronisation and/or sorting procedures. Here we introduce Nanotiming, a single-molecule, nanopore sequencing-based method producing high-resolution, telomere-to-telomere replication timing (RT) profiles of eukaryotic genomes by interrogating changes in intracellular dTTP concentration during S phase through competition with its analogue bromodeoxyuridine triphosphate (BrdUTP) for incorporation into replicating DNA. This solely demands the labelling of asynchronously growing cells with an innocuous dose of BrdU during one doubling time followed by BrdU quantification along nanopore reads. We demonstrate in *S. cerevisiae* model eukaryote that Nanotiming reproduces RT profiles generated by reference methods both in wild-type and mutant cells inactivated for known RT determinants. Nanotiming is simple, accurate, inexpensive, amenable to large-scale analyses, and has the unique ability to access RT of individual telomeres, revealing that Rif1 iconic telomere regulator selectively delays replication of telomeres associated with specific subtelomeric elements.

Eukaryotic chromosomes are duplicated according to a defined temporal sequence referred to as the replication timing (RT) programme[1,2], which plays a major role in chromatin structure maintenance and is associated with spatial organisation and transcriptional activity of the genome[1]. Reference techniques to profile RT in eukaryotes monitor either DNA copy number changes or the incorporation of non-canonical nucleotides during the S phase[3]. Most require cell synchronisation or cell sorting into one or multiple fractions of the S phase (sort-seq and Repli-seq methods, respectively)[3], which greatly complicates sample preparation. Moreover, cell synchronisation procedures may interfere with the DNA replication process (e.g., refs. 4–7), while inaccurate sorting gate positioning may distort RT profiles[3]. The marker frequency analysis (MFA)-seq approach directly assays the copy number of DNA sequences in proliferating cells[3], but is limited in resolution by the proportion of actively replicating cells in the population under scrutiny[8]. Current RT mapping techniques also remain expensive when adding up the costs of flow cytometry and next-generation sequencing. Altogether, these technical and financial constraints have precluded a more widespread use of RT profiling.

We and others have recently developed pipelines to detect the thymidine analogue 5-bromo-2′-deoxyuridine (BrdU) incorporated into replicating DNA using nanopore sequencing[9–13]. Since it has been observed in various eukaryotes, from yeast to mammalian cells, that dTTP pool size increases during S phase (e.g., refs. 14–21), we thought

[1]IBENS, Département de biologie, École normale supérieure, Université PSL, CNRS, INSERM, 75005 Paris, France. [2]Sorbonne Université, Collège Doctoral, 75005 Paris, France. [3]Laboratoire de Biologie et Modélisation de la Cellule, École Normale Supérieure de Lyon, CNRS, UMR5239, INSERM, U1293, Université Claude Bernard Lyon 1, 46 allée d'Italie, 69364 Lyon, France. [4]CNRS, ENS de Lyon, LPENSL, UMR5672, 69342 Lyon, cedex 07, France. [5]Present address: Bertrand Theulot, Department of Biology, New York University, 100 Washington Square East, New York, NY 10003, USA. [6]Present address: Jean-Michel Arbona, IBDM, UMR7288, Case 907, Parc Scientifique de Luminy, 13288 Marseille, Cedex 9, France. ✉e-mail: letallec@bio.ens.psl.eu

of using the BrdU content of DNA molecules from cells labelled with this analogue as a proxy for RT based on the reasoning that changes in BrdU level should reflect the reduction in the intracellular ratio of BrdUTP to dTTP as S phase unfolds. In other words, BrdU incorporation is expected to be lower in late than in early replicating regions. Here, we demonstrate in *Saccharomyces cerevisiae* that labelling asynchronously growing cells with BrdU for one doubling time followed by BrdU quantification along nanopore reads of genomic DNA is sufficient to produce a high-resolution RT profile of the budding yeast genome. This approach, which we call Nanotiming, eliminates the need for cell sorting or synchronisation to generate detailed RT maps, with a cost reduced to US$70 per profile when 24 multiplexed samples are sequenced simultaneously. It also leverages the possibility of unambiguously aligning long nanopore reads at highly repeated sequences to provide complete genomic RT profiles, from telomere to telomere. Notably, Nanotiming reveals that yeast telomere regulator Rif1[22] does not delay replication of all telomeres, as previously thought, but only of those associated with specific subtelomeric motifs. Furthermore, thanks to Nanotiming's unique ability to resolve RT on individual molecules, we can now investigate previously described connections between telomere length and RT[6,23] at the single-telomere level.

## Results

### Nanotiming foundations and proof of concept

We turned to *S. cerevisiae*, a widely studied model organism, to understand eukaryotic DNA replication to evaluate if RT can be accessed via BrdU-mediated capture of S phase dTTP pool expansion. Since fungi naturally lack a thymidine salvage pathway needed for the uptake of extracellular thymidine and its analogues, we used the BT1 strain, which is amenable to highly efficient BrdU incorporation thanks to the joint expression of human equilibrative nucleoside transporter 1 (hENT1) and Herpes simplex virus thymidine kinase (hsvTK) converting BrdU into cell-usable BrdU monophosphate[13]. We first conducted a pilot experiment aiming at defining the optimal BrdU concentration to seize the estimated 8-fold increase in dTTPs during the yeast S phase[17]. An intracellular dTTP concentration of ≈12 μM at the start of the S phase in *S. cerevisiae*[17] suggests that a BrdU dose in the same range should be used. However, because the rates of BrdU import and conversion into nucleotides are not known in BT1, we explored a broad variety of labelling BrdU concentrations, from 5 μM to 1 mM. Cells were incubated with BrdU for one complete doubling time so that the whole population would have undergone genome replication (Fig. 1a). BrdU detection was performed in Oxford Nanopore Technologies (ONT) reads of genomic DNA with our in-house BrdU basecalling model[13]. Strong variations in BrdU content (i.e., in the proportion of BrdU-substituted thymidine sites) were visible between and within nanopore reads from cells labelled with 5–20 μM BrdU (Fig. 1b), indicating that intracellular BrdUTP and dTTP concentrations were of the same order of magnitude in those conditions. Conversely, for 1 mM BrdU labelling dose, a majority of reads displayed a uniformly high BrdU level (Fig. 1b), pointing to a large excess of BrdUTPs over dTTPs, while "100 μM" reads had an intermediate profile. These results, therefore, show that S phase dTTP pool fluctuations in the S phase can be indirectly interrogated by labelling cells with a BrdU concentration in the 5–20 μM range. To determine if changes in BrdU level along nanopore reads mirror the decreasing intracellular BrdUTP to dTTP ratio and echo cell progression in the S phase, we next compared population-averaged BrdU values in 1 kb bins of the yeast genome with the relative copy number obtained by sort-seq (i.e., the ratio between the number of copies of a given locus in replicating and non-replicating cells estimated through NGS sequencing of genomic DNA from S- and G1/G2-sorted cells), which parallels the replication time of a locus in the cell population[8] (Fig. 1c). BrdU-free bins of nanopore reads, representing about half of the data (55.6, 51.1 and 52.2% for 5, 10 and 20 μM, respectively) and corresponding either to parental DNA or

to DNA replicated before medium supplementation with BrdU, were filtered out. As expected, mean BrdU content and relative copy number were positively correlated (Fig. 1c). Correlation was very strong for 5–20 μM BrdU doses (Spearman's rank correlation coefficient from 0.89 to 0.94), declined for 100 μM and further dropped for 1 mM (Spearman's rank correlation coefficient of 0.68 and 0.36, respectively), in line with our previous conclusion from individual reads that 5–20 μM BrdU concentrations can optimally catch the rise in dTTP levels during S phase. Importantly, for this BrdU concentration range, mean BrdU content had a near-linear relationship with relative copy number (Fig. 1c), implying that plots of mean BrdU content along the yeast genome are actual RT profiles.

To better understand the relationship between mean BrdU content and time, we extracted from 5, 10 and 20 μM BrdU data points in Fig. 1c the dynamics of dTTP concentration during the S phase. Indeed, considering that the mean BrdU content (MBC) reflects the intracellular BrdUTP (B) to dTTP (T) ratio (MBC = B/(B + T)), the dTTP pool can be estimated by inverting the terms of the equation (T = B*(1/MBC-1)). Satisfactorily, all three concentrations recovered the exponentially-shaped dTTP level increase in the S phase seen in *S. cerevisiae*[17] (Supplementary Fig. 1). To further illustrate the mathematical relationship connecting dTTP level, mean BrdU content and time, the exponential function that best fitted the increase in dTTP concentration in S was in turn inserted in the MBC = B/(B + T) formula, recapitulating the observed, quasi-linear mean BrdU content decrease in the same time interval (Supplementary Fig. 1). To directly demonstrate that the mean BrdU content declines during S phase as a result of dTTP pool expansion, we next turned to the MCM869 strain, inactivated for the thymidylate synthase-encoding *CDC21* gene required for de novo dTMP biosynthesis and therefore completely deprived of endogenous dTTPs[24]. To grow, MCM869 exclusively relies on exogenous thymidine or its analogues, which are exploitable owing to a reconstructed thymidine salvage pathway similar to that of BT1[13,24]. Because of their lack of dTTP level increase in the S phase, BrdU-labelled MCM869 yeasts are predicted to yield nanopore reads with homogenous BrdU levels. Accordingly, the mean BrdU content computed from reads of genomic DNA of MCM869 cells grown with various mixtures of BrdU and thymidine in the culture medium remained constant throughout S phase, with a value that simply echoed the extracellular proportion of BrdU (Supplementary Fig. 2). Altogether, these results certify that the near-linear mean BrdU content decrease measured in BT1 cells as they progress in S phase stems from the exponential increase of their endogenous dTTP pool in the meantime.

We decided to retain the 5 μM BrdU dose for the remaining experiments since the labelling of BT1 cells for one doubling time with 10 or 20 μM BrdU tended to alter their progression in S phase (Fig. 1d). Juxtaposing sort-seq relative copy number and "5 μM" mean BrdU content profiles along yeast chromosomes visually confirmed their resemblance, with both profiles being virtually superimposed (Fig. 2a and Supplementary Fig. 3). Remarkably, resolution of the mean BrdU content profile surpassed that of relative copy number profiling by MFA-seq (Fig. 2b and Supplementary Fig. 4), the only other RT mapping technique without cell sorting or synchronisation. Moreover, the mean BrdU content profile extended over the complete length of BT1 telomere-to-telomere assembly (see Methods). In sharp contrast, distal chromosomal regions were missing on sort-seq and MFA-seq profiles, which aggregate short Illumina reads that cannot be uniquely mapped in repeated DNA sequences (Fig. 2a, b and Supplementary Figs. 3, 4). Similarly, RT of the repetitive rDNA locus, which was previously inaccessible, could now be retrieved (Fig. 2a, b). To evaluate reproducibility, we generated five additional, independent mean BrdU content profiles of the *S. cerevisiae* genome (Supplementary Fig. 5). We recovered Spearman's rank correlation coefficients ranging from 0.96 to 0.97 between all six replicates (Supplementary Fig. 6), highlighting the robustness of our approach.

 

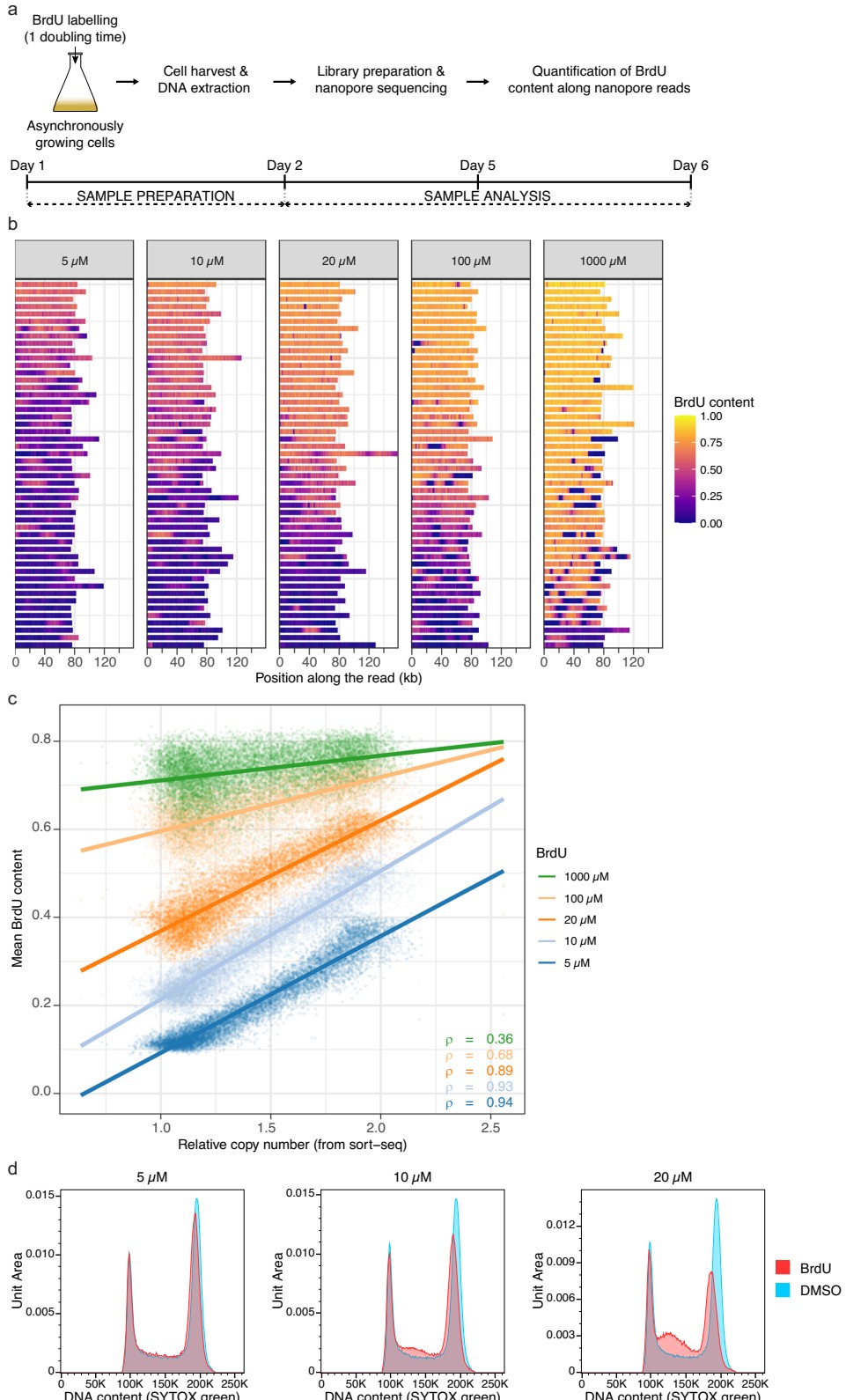

Finally, we investigated whether our method was able to detect RT alterations by profiling mean BrdU content along the genome of mutant yeasts where key RT regulators, namely Ctf19, Rif1 and Fkh1, had been inactivated. Ctf19 and Rif1 control the early replication of centromeres[25] and late replication of telomeres[6,7,26–28], respectively, while Fkh1 governs the early firing of a subset of yeast origins[29]. Furthermore, Nanotiming was applied to cells inactivated for the Ku complex (through deletion of the *YKU70* gene), which exhibit early-replicating chromosome ends attributed to telomere shortening[5,6]. All mutant RT profiles were made in triplicate from independent cell cultures. As anticipated, we were able to retrieve late pericentric replication in the absence of Ctf19 (Fig. 2c and Supplementary Fig. 7) and telomeric switch to early replication both in the *rif1Δ* and *yku70Δ* strains (Fig. 2d, e and Supplementary Figs. 8, 9). In addition, we found

**Fig. 1 | Accessing *S. cerevisiae* genome RT by BrdU-mediated capture of dTTP pool expansion during S phase. a** Scheme of the protocol for BrdU labelling of replicating DNA in BT1 cells followed by BrdU quantification in nanopore reads of genomic DNA. The typical timeline is indicated. See text for BrdU concentrations used in this study. **b** BrdU content fluctuations along nanopore reads of genomic DNA from BT1 cells labelled for one doubling time with BrdU. The BrdU content in 1 kb bins along individual nanopore reads, corresponding to the fraction of the thymidine sites of these bins that incorporated a BrdU (i.e., to their BrdU/(BrdU +Thymidine) ratio), is represented as a 1D heatmap. Reads were randomly selected and are organised according to their median BrdU content (from top to bottom, median BrdU content from highest to lowest); BrdU-free reads (median BrdU content ≤ 0.02) corresponding to parental DNA were filtered out. The BrdU dose used for cell labelling is indicated on top of each panel. **c** Comparison between relative copy number established by sort-seq in 1 kb bins of BT1 genome and mean BrdU content in the cognate bins computed from nanopore reads of genomic DNA of cells labelled with the indicated BrdU dose for one doubling time. A linear regression model (coloured line) and Spearman's rank correlation coefficient (ρ) between mean BrdU content and relative copy number are given for each labelling BrdU concentration. **d** Labelling of BT1 cells for one doubling time with 5 μM BrdU does not impact S phase progression. Representative analysis of BT1 cell cycle after exposure to DMSO or 5, 10, or 20 μM BrdU for one doubling time. DNA content was analysed by flow cytometry after DNA staining with SYTOX Green. The BrdU dose is indicated above each panel. This experiment was performed three times independently with similar results.

that Rif1 regulates RT at several chromosome-internal origins as well as the RT of the rDNA locus, as already reported[7,30–32] (Fig. 2d and Supplementary Fig. 8). The slightly later RT of this locus in *yku70Δ* cells (Fig. 2e) points to a potential role for the Ku complex in stimulating origin firing in the rDNA, unless it merely reflects an increased competition with telomeric origins for limiting DNA replication initiation factors. We did detect a delayed activation of a large number of origins in *fkh1Δ* mutant cells but also noted that a subset of origins fires earlier without Fkh1, in agreement with previous observations[29] (Fig. 2f and Supplementary Fig. 10). Taken together, these results indicate that our technique can reveal both global or more subtle modifications of the RT programme. Noticeably, for *ctf19Δ* and *rif1Δ* mutants, genomic profiles of mean BrdU content and relative copy number from published sort-seq experiments[7,25] almost fully overlapped (Spearman's rank correlation coefficient of ≈ 0.92 for *ctf19Δ* and ≈ 0.82 for *rif1Δ*, Supplementary Figs. 11–13), as they did for wild-type cells (Fig. 2a and Supplementary Fig. 3).

In conclusion, our study demonstrates that it is possible to build high-resolution RT profiles of a eukaryotic genome by measuring changes in intracellular dTTP concentration during the S phase through competition with BrdUTP for incorporation into DNA. We named this approach "Nanotiming".

## Nanotiming reveals that Rif1 selectively delays replication of a subset of yeast telomeres

In the yeasts of *S. cerevisiae* and *S. pombe*, it is widely accepted that Rif1 controls the late RT of telomeres[22]. Rif1 is recruited therein by a telomere-binding protein (Rap1 and Taz1 in budding and fission yeast, respectively) and, in turn, recruits the PP1 phosphatase to replication origins to counteract activating phosphorylation of the replicative helicase[26–28]. Intriguingly, when we analysed *rif1Δ* telomere-to-telomere RT profile, we observed that although most chromosome ends shifted to earlier RT in accordance with previous sort-seq results[7], the amplitude of RT alterations vastly differed between telomeres (Fig. 3a and Supplementary Fig. 8). Since *S. cerevisiae* chromosome extremities have distinct subtelomeric regions comprising a single X and none to multiple Y' repetitive elements, each containing a replication origin[33], we decided to study RT changes at telomeres in *rif1Δ* cells in light of their subtelomeric structure, which was fully accessible to long nanopore reads. Strikingly, RT modification was evident at all 20 XY' telomeres but was more tenuous, or even not visible, at telomeres only containing an X element (Fig. 3 and Supplementary Fig. 14). In contrast, both X and XY' telomeres replicated much earlier in *yku70Δ* strain (Supplementary Figs. 15, 16). Moreover, Nanotiming profiles exposed that the slightly advanced RT at certain X telomeres in *rif1Δ* cells systematically stemmed from the upregulation of a chromosome-internal origin located close to the telomere, and not because of an earlier firing of the telomeric X origin (Fig. 3c and Supplementary Fig. 14); this was especially the case for TEL06R and TEL15L, two model telomeres that have recurrently been shown to replicate earlier in the absence of Rif1[27,28]. Conversely, RT change at XY'

telomeres always emanated from chromosome extremity (Fig. 3c and Supplementary Fig. 14), which was the case for both X and XY' telomeres in the *yku70Δ* mutant (Supplementary Fig. 16). We conclude that Rif1 directly regulates the RT of XY' telomeres alone.

## Nanotiming accesses the RT of individual telomeres

Telomeres are heterogeneous in length within a defined range (300 ± 75 bp of TG$_{1-3}$ repeats in budding yeast[33]). Artificially shortened telomeres replicate early in *S. cerevisiae*[23] and the short telomere length in *yku70* mutants is held responsible for their early RT[6], suggesting a link between RT and telomere length homoeostasis. Interestingly, long-read sequencing has recently been used to estimate the length of individual telomeres in a wide range of organisms (e.g., refs. 34–40). Nanotiming is, therefore, uniquely positioned to study the relationship between telomere length and RT in yeast. We first validated our nanopore sequencing-based procedure to measure the length of telomeric repeats by (i) retrieving telomeres globally comprised between 200 and 400 bp (median length of ≈ 300 bp) in wild-type (plus *ctf19Δ* and *fkh1Δ*) cells, with few telomeres outside these bounds, (ii) detecting the well-characterised telomere elongation in the absence of Rif1[41] and (iii) precisely recovering the ≈ 150–200 bp decrease in telomere length expected upon Ku complex inactivation[42,43] (Fig. 4a); it was also ascertained by the high reproducibility between biological replicates. We then leveraged Nanotiming's capacity to access RT on individual molecules through BrdU content computation along a given nanopore read to compare telomere length and RT at the single-telomere level in wild-type, *rif1Δ*, *yku70Δ*, *ctf19Δ* and *fkh1Δ* cells (Fig. 4b–f, Supplementary Figs. 17–21 and Supplementary Data 1–3). Our analysis showed no particular association between both parameters whatever the strain and chromosome extremity under scrutiny, except perhaps for *yku70Δ* mutant where shorter X telomeres tended to replicate earlier (Spearman's rank correlation coefficients of ≈ − 0.2, Fig. 4d, Supplementary Fig. 19 and Supplementary Data 2). Telomeres predominantly exhibited a late RT in wild-type, *ctf19Δ* and *fkh1Δ* cells except for a fraction that replicated more precociously, independently of their length (Fig. 4b, e, f and Supplementary Figs. 17, 20, 21). This fraction was logically higher for telomeres displaying an advanced RT at the population level (compare, for example, TEL07L and TEL07R in Figs. 3a and 4b). As expected, it dramatically expanded in *yku70Δ* and *rif1Δ* backgrounds – exclusively in the case of XY' telomeres for the latter –, once again, mostly irrespective of telomere length (Fig. 4c, d and Supplementary Figs. 18, 19).

## Nanotiming, a cost-effective RT profiling method suitable for large-scale analyses

In order to evaluate Nanotiming RT profiles' quality as a function of sequencing depth, subsampling analyses were performed by randomly selecting a progressively smaller number of reads from the complete dataset used to compute the Nanotiming profile of *S. cerevisiae* genome presented in Fig. 2a. The resulting RT maps were then compared

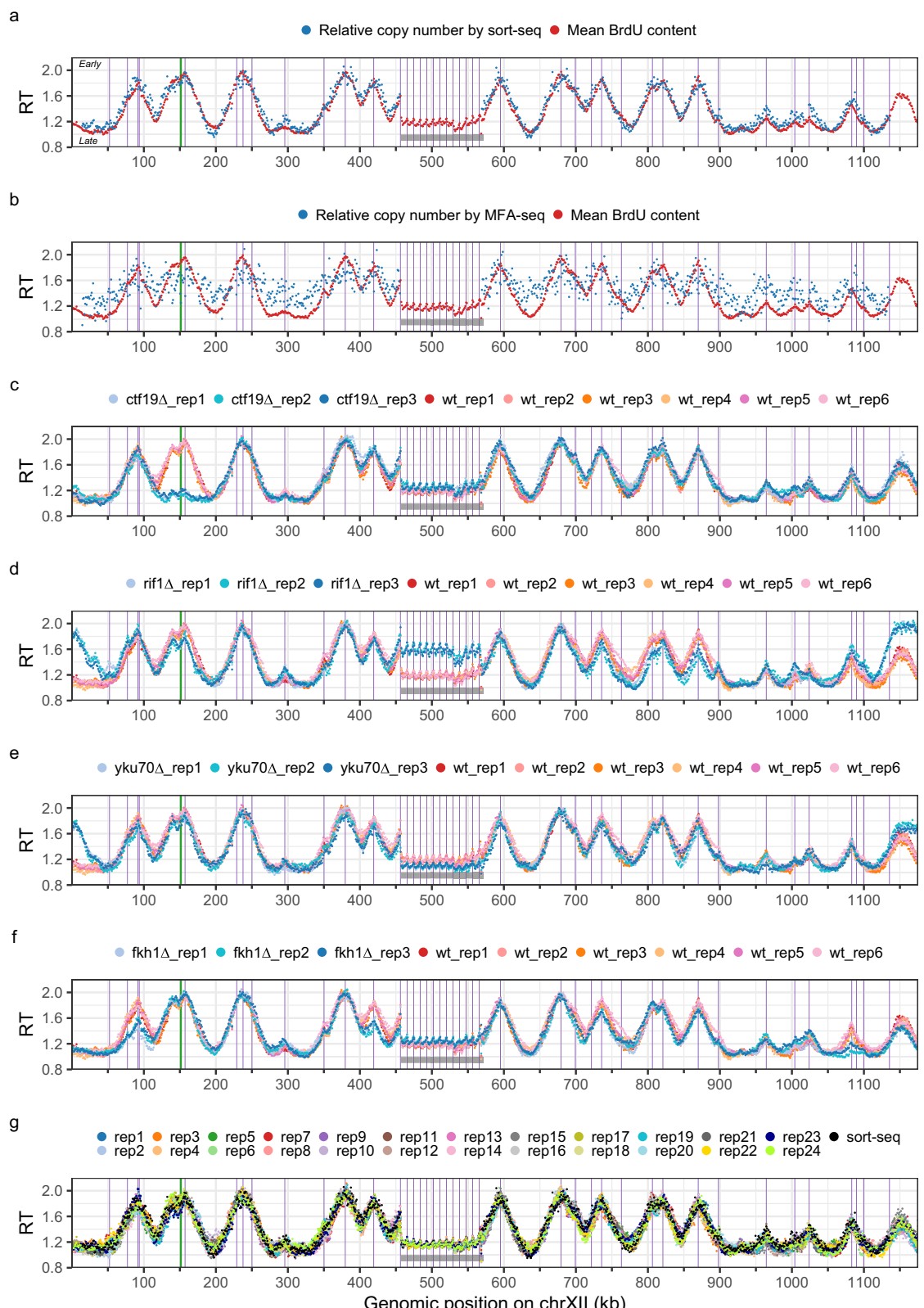

with sort-seq. A plateau with Spearman's rank correlation coefficient values greater than 0.9 was reached for profiles built from only ≈ 50,000 nanopore reads (average read length of ≈ 20 kb), equivalent to a ≈ 100x coverage or ≈ 1 Gb of mapped DNA (Supplementary Fig. 22). This indicates that small amounts of sequencing data are sufficient to produce reliable RT profiles by Nanotiming, probably because of low intrinsic noise (Supplementary Fig. 23). The advertised 100–200 Gb output of ONT PromethION flow cell together with multiplex sequencing, therefore, make it possible to generate dozens of RT profiles of the yeast genome in a single PromethION run, drastically decreasing sequencing costs per sample. Accordingly, we performed a multiplexed experiment with 24 barcoded samples prepared from the same BT1 BrdU-labelled genomic DNA, which resulted in 24 near-identical Nanotiming profiles all closely resembling sort-seq

**Fig. 2 | RT profiles of *S. cerevisiae* chromosome XII using Nanotiming. a** Mean BrdU content and sort-seq relative copy number profiles (reads of genomic DNA from BT1 cells). **b** Mean BrdU content and MFA-seq relative copy number profiles (data from BT1 cells and from ref. 8, respectively). **c–f** Mean BrdU content profile in wild-type (wt) and *ctf19Δ* (**c**), *rif1Δ* (**d**), *yku70Δ* (**e**) and *fkh1Δ* (**f**) BT1 cells. **g** Mean BrdU content profiles from a multiplexed PromethION run with 24 barcoded samples of BT1 BrdU-labelled DNA; the corresponding sort-seq relative copy number profile is also shown. **a–g** Mean BrdU content and relative copy number were computed in 1 kb bins; data were rescaled between 1 (end of S phase) and 2 (start of S phase) for comparison (see "Methods"). Mean BrdU content profiles were calculated using nanopore reads of genomic DNA from cells labelled with 5 μM BrdU for one doubling time; six and three biological replicates, corresponding to

independent cell cultures, are presented in **c–f** for wild-type and mutant BT1 cells, respectively; BT1 wt_rep1 mean BrdU content profile is shown in (**a**) and (**b**); the 24 samples in (**g**) originate from the same BT1 BrdU-labelled genomic DNA. Purple and green vertical lines, positions of confirmed *S. cerevisiae* replication origins and of centromeres, respectively; grey box, rDNA. BT1 assembly rDNA locus comprises 12 copies of the 9.1 kb rDNA repeat, with some discontinuities due to contig assembly and scaffolding issues (see Methods); please note that, although reads with "external" rDNA repeats flanked by sequences located upstream or downstream of the rDNA locus, allowing unambiguous mapping, contribute to the BrdU signal, most of the rDNA reads only contain repeats and are therefore randomly mapped on the 12 rDNA copies, giving rise to an average RT profile of the rDNA locus. RT, replication timing; rep, replicate.

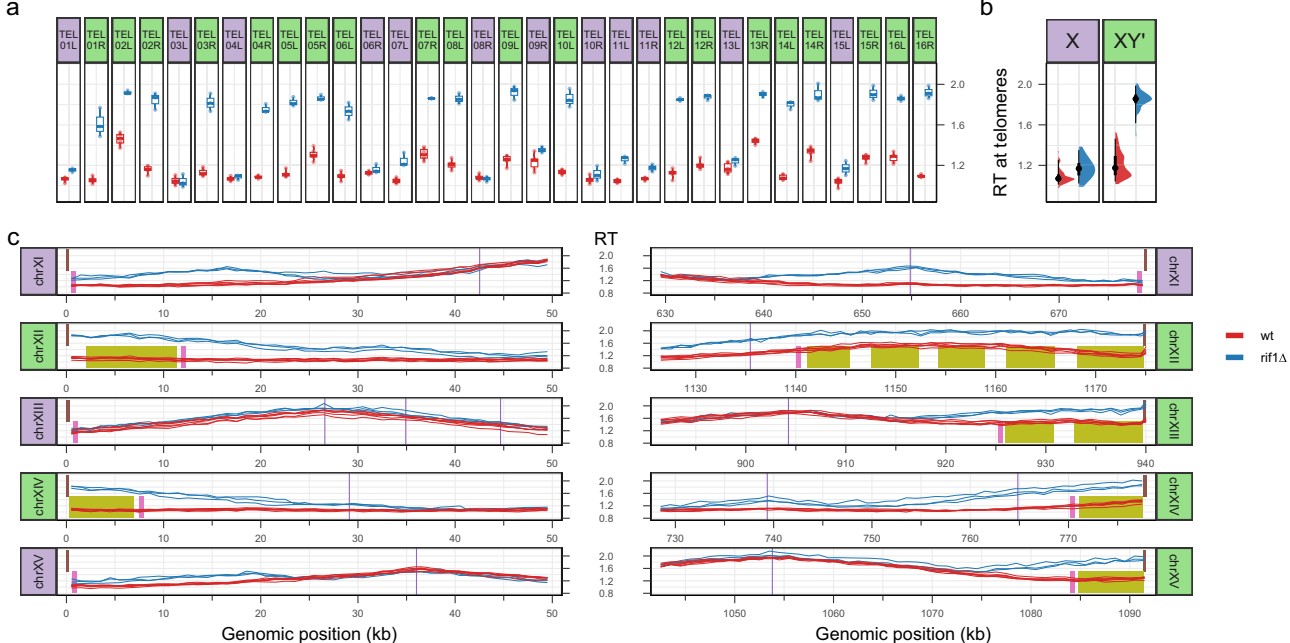

**Fig. 3 | Rif1 controls the late RT of *S. cerevisiae* XY', but not X, telomeres. a** RT at individual telomeres in wild-type (wt) and *rif1Δ* BT1 cells. Dot, mean BrdU content in a given sample of either the first (for left (L) telomeres) or last (for right (R) telomeres) 1 kb bin of the cognate chromosome adjacent to the terminal telomeric repeats, with purple and green backgrounds distinguishing X and XY' telomeres, respectively; box, 50% interval; thick horizontal line, median value; whiskers, extreme values. $n_{wt} = 6$ mean BrdU content values from independent cell cultures, $n_{rif1Δ} = 3$. **b** RT at X and XY' telomeres in wt and *rif1Δ* cells. Half-eye plots aggregate values at individual telomeres presented in (**a**) according to their X/XY' status. For X telomeres, $n_{wt} = 72$ mean BrdU content values, $n_{rif1Δ} = 36$; for XY' telomeres, $n_{wt} = 120$, $n_{rif1Δ} = 60$. Black dot, median RT; thick and thin black vertical lines, 50%

and 95% intervals, respectively. **c**, Mean BrdU content profiles over 50 kb of the left and right extremities of chromosomes (chr) XI-XV in wt and *rif1Δ* BT1 cells. Purple vertical lines, positions of confirmed *S. cerevisiae* replication origins; brown, pink, and olive boxes, TG_{1-3} repeats, subtelomeric X and Y' elements, respectively. Purple and green backgrounds distinguish chromosome extremities with X and XY' telomeres, respectively. Six and three biological replicates, corresponding to independent cell cultures, are presented for wt and *rif1Δ* BT1 cells, respectively. **a–c** wt and *rif1Δ* samples are in red and blue colours, respectively. Mean BrdU content was computed in 1 kb bins from nanopore reads of genomic DNA of cells labelled with 5 μM BrdU for one doubling time and rescaled as in Fig. 2. RT, replication timing.

(Spearman's rank correlation coefficients for comparisons between replicates and with sort-seq systematically over 0.9, Fig. 2g and Supplementary Figs. 24, 25), for as little as US$70 per genome including consumables and reagents (Supplementary Table 1). In fact, our PromethION run generated ≈ 80 Gb of sequenced DNA, that is 3.3 Gb/sample, far above the 1 Gb necessary to compute a sound RT map. It is thus reasonable to envision the multiplexing of up to 96 samples on a PromethION flow cell, which is the current limit of ONT barcoding ligation kits, setting the price to just US$10 per genomic RT profile when using ONT's latest R10 chemistry (Supplementary Table 1).

## Discussion

Conventional RT studies track the doubling of copy numbers or follow the incorporation of thymidine analogues during DNA replication[3]. We

demonstrate here in *S. cerevisiae* that RT can additionally be profiled in eukaryotes by interrogating S phase dTTP pool increase via BrdU competitive incorporation into replicating DNA, a method that we refer to as Nanotiming. We show that the mean BrdU content along nanopore reads of genomic DNA from asynchronously growing cells labelled for one doubling time with this analogue has a positive, quasi-linear relationship with the relative copy number calculated by sort-seq, which has itself a negative linear relationship with the median replication time[8]. In line with our results, an inverse correlation between the average level of BrdU incorporation and the average replication time has been lately reported in yeast[9]. Simply put, computing the mean BrdU content along yeast chromosomes is equivalent to profiling RT, which is further illustrated by the near-perfect overlap between RT profiles of the yeast genome obtained by either

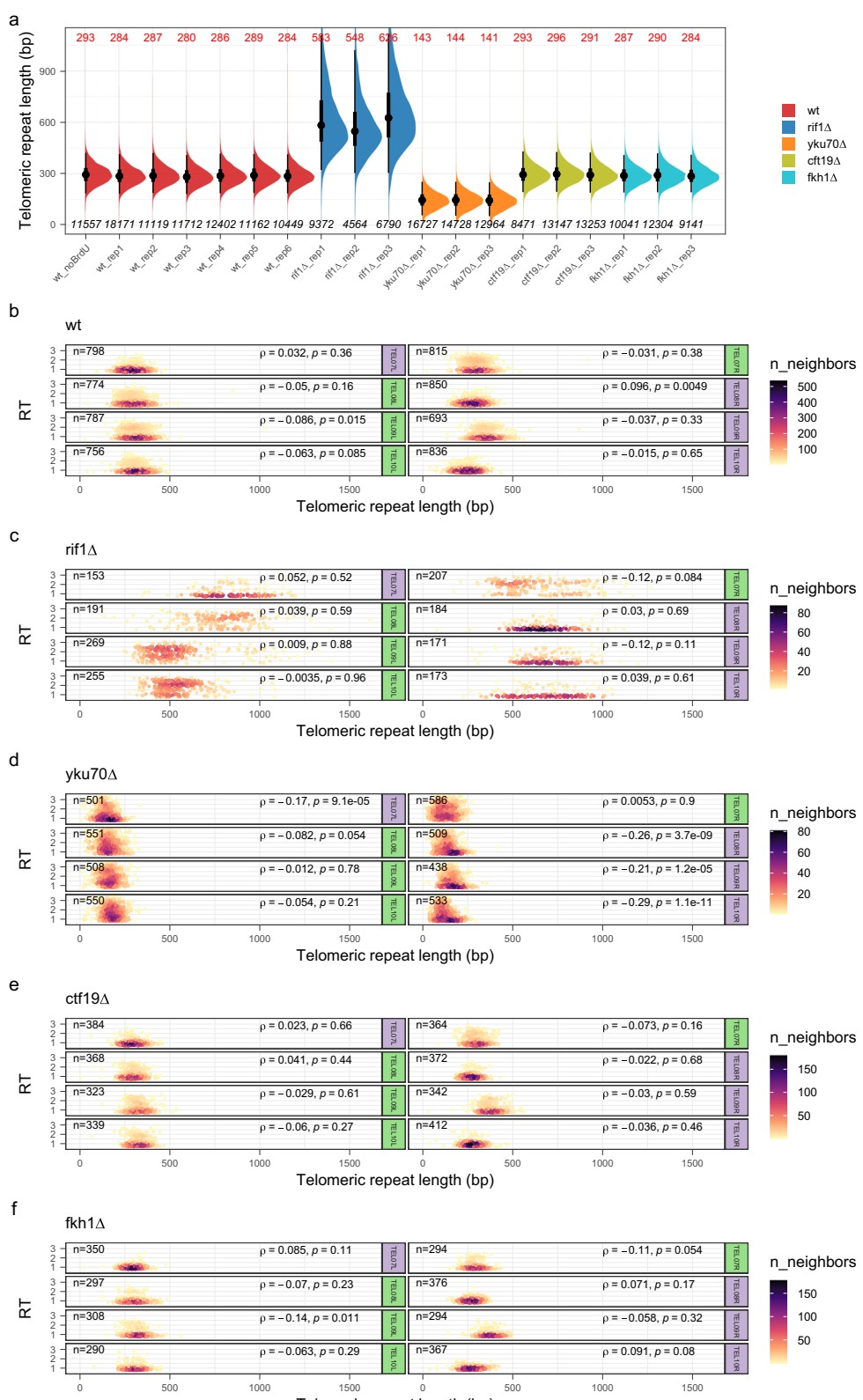

Nanotiming or sort-seq. The mean BrdU content profile actually displays sharper peaks as our technique is not constrained by the arbitrary delimitation of the G1/S border accompanying the sorting of S phase cells that tends to exclude very early-replicating regions. We note that the latest peaks are of generally smaller amplitude on Nanotiming than on sort-seq profiles, which likely reflects that the linear relationship between nanopore read mean BrdU content and

time is compromised when dTTP concentration becomes much larger than that of BrdUTP, as one might expect (see below). By matching the resolution of sort-seq without the need for cell sorting and by outperforming MFA-seq, Nanotiming is set to become the method of choice for temporal studies of DNA replication. Nanotiming is a robust methodology, as demonstrated by the similarity of all six independent RT profiles of *S. cerevisiae* genome generated in our study, and is able

**Fig. 4 | Analysis of the relationship between telomere length and RT at the single-telomere level in wild-type, *rif1Δ*, *yku70Δ*, *ctf19Δ*, and *fkh1Δ* BT1 cells.**
**a** Telomere length in the indicated strain. Half-eye plots show the distribution of individual telomeric repeat lengths measured using nanopore reads of genomic DNA from independent cell cultures of the corresponding strains. Telomere length was concurrently estimated in unlabelled BT1 cells (wt_noBrdU sample) to ascertain that BrdU incorporation has no impact on telomere length estimation. TEL13R length was not determined in the *yku70Δ* mutant due to a missing Y' element at the right end of chromosome XIII compared to BT1 assembly, preventing proper read mapping and telomeric repeat length measurement. Black dot, median telomeric

repeat length, value indicated in red on top; thick and thin black vertical lines, 50% and 95% intervals, respectively; bottom, number of individual telomeric repeat length measurements. **b**–**f**, RT *versus* length of single telomeres plotted as a 2D density plot in the indicated strain for TEL07L/R, TEL08L/R, TEL09L/R, and TEL10L/R. Coefficient (ρ) and *p*-value of Spearman's correlation test (two-sided) between telomeric repeat length and RT as well as the number of measurements (n) are indicated. Purple and green backgrounds distinguish X and XY' telomeres, respectively. Mean BrdU content data were rescaled as in Fig. 2. RT, replication timing; rep, replicate; wt, wild-type.

to detect diverse alterations of the RT programme, as shown in *ctf19Δ*, *rif1Δ*, *yku70Δ* and *fkh1Δ* yeast mutants. Its simplicity means that numerous replicates can be produced to ensure reliable results. Moreover, those can be analysed all at once by performing multiplex sequencing, which greatly reduces costs – at least 24 yeast genomic RT profiles can be built using a single PromethION flow cell, for a unit price of just US$70. Lastly, Nanotiming takes less than a week (Fig. 1a) and can be fully performed in-house, as it only demands an affordable ONT device. In comparison, although the sort-seq procedure theoretically takes about the same amount of time as Nanotiming[44], the cell sorting and NGS sequencing, it requires are generally only accessible from facilities or companies, which often means a longer processing time.

In addition to yeast[17–20], several studies have documented that intracellular dTTP levels expand in the course of the S phase in mammalian cells (e.g., refs. [14–16,21]). It is thus tempting to speculate that dTTP increase during the S phase is a feature shared by all eukaryotes, especially considering that nucleotide biosynthesis pathways are evolutionarily conserved. Still, it is impossible to guarantee that dTTP pool kinetics throughout S phase in every eukaryotic cell will resemble the exponential increase in *S. cerevisiae* which results in a near-linear relationship between mean BrdU content and replication time. It should, however, be noted that even in the case of a non-linear relationship between both parameters, accessing RT remains possible as long as this relationship is strictly monotonic (i.e., if the mean BrdU content is always decreasing over time, regardless of how it decreases) since a strictly monotonic function can be inverted mathematically. We are, therefore, confident that Nanotiming can be applied to a broad range of eukaryotic organisms. By its very nature, though, this method will not work under conditions preventing dTTP pool change in the S phase, whether due to mutations in dNTP metabolism genes or the use of certain drugs.

Detecting BrdU in DNA is now a straightforward process thanks to the recent development of several BrdU basecallers using long-read sequencing technologies from either ONT[9–13] or PacBio[45], working on BrdU-labelled DNA from yeasts[9,10,12,13] to *Plasmodium falciparum*[46,47] to mammals[11,45], and accessible from public repositories. Consequently, the main challenge for Nanotiming implementation is to determine the amount of BrdU nucleosides to supplement the growth medium so that these will be converted into the appropriate concentration of intracellular BrdUTP to probe dTTP fluctuations. If the BrdU dose is too high, it will result in almost exclusive incorporation of BrdUTPs into newly-synthesised DNA at any point in S and mask the rise in dTTP levels, while an insufficient amount of BrdU will cause BrdUTPs to be poorly integrated into DNA during replication and thus less detectable. We show in yeast that solid RT analyses can be performed using a relatively broad range of BrdU concentrations, indicating that there is some flexibility in labelling conditions. As a general guideline, we recommend selecting a BrdU dose resulting in the broadest distribution of BrdU contents along nanopore reads, as is the case with 5–20 μM concentrations for BT1 cells (Fig. 1c). Nevertheless, we recognise that Nanotiming needs careful tuning. This includes verifying the innocuity of the BrdU dose used for cell labelling, although we noted that RT profiles generated from cells treated with 10 or 20 μM

BrdU, which display altered S phase progression (Fig. 1d), are in fact remarkably similar to the "5 μM" and sort-seq profiles (Supplementary Fig. 26). Nanotiming also requires a functional thymidine salvage pathway, which is present in most eukaryotes, including metazoans and plants, but is lacking in fungi and notably in the yeasts *S. cerevisiae* and *S. pombe* where, however, it is routinely reconstituted through the combined expression of TK and a nucleoside transporter[48,49]. This is the case with the BT1 strain, which contains the pBL-hsvTK$_{CO}$-hENT1$_{CO}$ construct for efficient BrdU incorporation[13], illustrating that applying Nanotiming to naturally TK-less organisms is feasible. Above all, once set up, Nanotiming is certainly the easiest high-resolution RT profiling method currently available. Sample preparation is simple, and after BrdU basecalling, mean BrdU content calculation (available on GitHub at https://github.com/LacroixLaurent/NanoTiming) is not computationally demanding. For maximum versatility, we provide future users of Nanotiming in budding yeast with a set of pBL vectors with 4 selectable markers, namely *URA3*, *TRP1*, *HIS3* and *AUR1-C*.

Telomere length heterogeneity at a given chromosome end, together with interchromosomal variability in the number of telomeric repeats and in subtelomeric regions make each telomere unique, yet replication at individual telomeres has remained largely inaccessible due to near-impossible telomeric read mapping to specific chromosome extremities with conventional sequencing. This barrier is now lifted with Nanotiming, which makes use of the long nanopore reads to reveal that Rif1, presented as the main regulator of telomeric RT in *S. cerevisiae*[22], only governs the RT of XY' telomeres; most X telomeres do replicate slightly earlier in the absence of Rif1, but only because of the upregulation of chromosome-internal, telomere-proximal origins. This previously went unnoticed presumably because of local, primer-based RT estimations unable to discriminate forks coming from the centromeric or telomeric side and, more recently, due to incomplete telomeric regions on RT profiles built from microarrays or short-read sequencing. Incidentally, several studies have reported telomere-specific regulations depending on subtelomere X/Y' composition (e.g., refs. [34,50,51]); whether those are connected remains to be determined. We note that X telomeres are globally replicated later than XY' telomeres in wild-type cells, in line with old observations that TEL05R replicates earlier when the Y' element is present than when it is not[52]. X telomere RT seems essentially dictated by the distance to, and firing time of, the closest active chromosome-internal origin (compare for instance TEL11L and TEL13L in Fig. 3c), whereas differences in RT at XY' telomeres are compatible with disparate efficiencies of Y' origins, primarily controlled by Rif1, as exemplified for chromosome XIV extremities (Fig. 3c). We failed to detect any peculiar association between RT and telomere length other than a weak trend for shorter X telomeres to replicate earlier in *yku70Δ* cells when directly comparing RT and telomere length at the single-telomere level. In contrast with previous studies suggesting that telomere shortening leads to earlier replication[6,23], we also observed that XY' telomeres are replicated even earlier than X telomeres in *yku70Δ* cells (Supplementary Fig. 15) despite both being equally short (Supplementary Data 2). Altogether, our data therefore suggest that (i) the regulation of telomeric RT and that of telomere length are separate mechanisms, as previously proposed[31,53]; (ii) the presence of subtelomeric Y' element(s), likely via

the accompanying replication origin(s), is the most important feature influencing yeast telomeric RT. The relief of a Ku-dependent, cumulative inhibitory effect on X and Y' origins would then explain the earlier replication of XY' telomeres compared to X ones in the *yku70Δ* mutant. On another note, we observed systematic rearrangements affecting Y' elements upon Ku inactivation (7 independent clones tested; see also Supplementary Fig. 16), possibly because very short telomeres in *yku70Δ* cells are unstable and hence more exposed to recombination between subtelomeres. Interestingly, single-molecule RT analysis shows that a fraction of telomeres are replicated early in wild-type, *ctf19Δ* and *fkh1Δ* cells and that telomeres (XY' telomeres) are not systematically replicated early in the absence of Ku70 (Rif1) but rather have a greater probability of being so. This result is fully consistent with *S. cerevisiae* genome replication being a stochastic process, with large cell-to-cell variability in origin choice and activation time[54].

In summary, Nanotiming provides simple, low-cost, high-quality, truly genome-wide RT profiling, making temporal replication studies accessible to all laboratories, and opening the way to large-scale RT analyses. Its unique assets help open the black box of replication at individual telomeres, a crucial step forward to elucidate telomere replication strategy, which is itself essential to preserve chromosome extremities and safeguard genome integrity[55].

## Methods

### Yeast strains and growth conditions

Strains used in this study are BT1 (*MATa trp1-1 leu2-3,112 his3-11,15 bar1::LEU2 ura3-1::URA3-GPD-hsvTK_{CO}-ADH-hENT1_{CO}(5x)*)[13], BT21 (BT1 *ctf19::KANMX*), BT22 (BT1 *rif1::KANMX*), BT23 (BT1 *yku70::NATMX*) and BT24 (BT1 *fkh1::KANMX*); all are isogenic to W303. Standard yeast genetic techniques and media were used[56]. Cells were grown at 30 °C in a YPD medium (MP Biomedicals). Biological replicates correspond to independent cultures started from the same strain.

### Plasmids

pBL-HIS3-hsvTK_{CO}-hENT1_{CO}, pBL-TRP1-hsvTK_{CO}-hENT1_{CO} and pBL-AUR1C-hsvTK_{CO}-hENT1_{CO} plasmids are derivatives of p403-BrdU-Inc (Addgene plasmid # 71789; http://n2t.net/addgene:71789; RRID:Addgene_71789), p404-BrdU-Inc (Addgene plasmid # 71790; http://n2t.net/addgene:71790; RRID:Addgene_71790) and p306-BrdU-Inc (Addgene plasmid # 71792; http://n2t.net/addgene:71792; RRID:Addgene_71792) vectors, respectively (gifts from Oscar Aparicio[57]), in which *hsvTK* and *hENT1* genes were replaced by versions codon-optimised for expression in yeast (see ref. 13 for details; *URA3* marker of p306-BrdU-Inc was additionally replaced by an AUR1C cassette to create pBL-AUR1C-hsvTK_{CO}-hENT1_{CO}). All three plasmids, alongside pBL-hsvTK_{CO}-hENT1_{CO}[13], as well as detailed cloning procedures, are available upon request.

### BrdU labelling

BT1 cells and derivatives were grown overnight in YPD, diluted in fresh medium at a 600 nm optical density ($OD_{600}$) of ≈ 0.1 and labelled with BrdU after reaching $OD_{600}$ ≈ 0.8 for one $OD_{600}$ doubling (i.e., for one doubling time, typically 90 min) (see "Results" section for BrdU concentrations used). Cells were then pelleted and washed with water before DNA extraction as in our previous work[58].

### Analysis of BrdU labelling impact on BT1 cell cycle by flow cytometry

BT1 cells labelled with various BrdU doses as described above, as well as unlabelled cells cultured in the same conditions were fixed in 70% ethanol. Fixed cells were prepared for flow cytometry analysis as described in the "Sort-seq" section. Samples were analysed using a BD FACSMelody™ Cell Sorter. Data were collected with BD FACSChorus v1.3.2 and processed with FlowJo

v10.9.0. The gating strategy is illustrated in Supplementary Fig. 27a.

### Library preparation, data acquisition, BrdU basecalling and read mapping

Library preparation, data acquisition, BrdU basecalling and read mapping were performed as in ref. 13 (megalodon v2.2.9, guppy v4.4.1 and minimap2 v2.24) with BrdU model from https://github.com/LacroixLaurent/NanoForkSpeed. BT1 genome (BT1multiUra.fa) used for the mapping was assembled as described in the "BT1 genome Assembly" section below. Detailed information for BrdU samples sequenced in this study are presented in Supplementary Data 4.

### BT1 genome assembly

Nanopore reads of genomic DNA from unlabelled BT1 cells (BT1_B0 sample from ref. 13) were basecalled using guppy (v6.6.2) in super accuracy mode. Prior to genome assembly, mitochondrial reads were filtered out of the total dataset through alignment on the sacCer3 reference genome with minimap2 (v2.24). Remaining nuclear reads were extracted from the complete read set based on this list using Seqtk (v1.3), and subsequently downsampled to 100x coverage using Rasusa (v0.7.0). Genome assembly and annotation were carried out in accordance with the methods from O'Donnell and colleagues[35] (Canu v2.2, Racon v1.5, Medaka v1.7.2, Pilon v1.23, Ragout v2.3, LRSDAY v1.7) with the following modifications: nanopore read polishing was performed using trimmed reads from Canu, as Porechop was not maintained at the time of BT1 genome assembly; the length_cutoff_for_completeness parameter of LRSDAY.13.Y_Prime_Element_Annotation.sh script for Y' element annotation was adjusted from 3500 to 1000 to account for increased sequence degeneration of some elements (which explains why certain Y' elements appear smaller than the 5.2 and 6.7 kb standard sizes[33] in Fig. 3c and Supplementary Figs. 14, 16). Contig correction with Pilon was performed using Illumina reads from sort-seq G2 fraction (see "Sort-seq" section). A separate mitochondrial genome assembly was constructed using the mitochondrial read dataset with Flye (v2.9). The rDNA locus and integrated copies of pBL-hsvTK_{CO}-hENT1_{CO} plasmid were defined by mapping the corresponding sequences from SGD (https://www.yeastgenome.org/, S288C_reference_genome_R64-3-1_20210421) and our own database, respectively, using minimap2. Complete assembly of budding yeast rDNA locus, composed of 100–200 tandem copies of a 9.1 kb unit, is impossible with the current length of nanopore libraries. BT1 genome rDNA locus comprises 12 rDNA repeats, with internal discontinuities and imperfect junctions with the rest of chromosome XII due to contig assembly and scaffolding issues. In order to annotate replication origins (known as autonomously replicating sequences, ARSs, in *S. cerevisiae*) in the BT1 genome, ARS positions were downloaded from OriDB[59] (http://cerevisiae.oridb.org/) as of December 2023. OriDB ARSs are classified into three categories, namely "confirmed", "likely" and "dubious", with only "confirmed" ARSs having a proper name; we, therefore, designed a unique naming scheme for both the "likely" and "dubious" categories based on the juxtaposition of the chromosome name and of the relative position of a given "likely" (or "dubious") ARS on this chromosome. Two ARSs, which were both named "ARS302" in OriDB were renamed as ARS302_1 and ARS302_2. DNA sequences corresponding to the defined ARSs were then extracted from sacCer1, the yeast reference genome version used in OriDB, and the resulting fasta file (AllARS_sacCer1_20231218.fa) was subsequently mapped onto the BT1 genome using bwa mem (v0.7.17-r1198-dirty). The output bam file was then sorted and indexed with samtools (v1.17) and converted to bed using bedtools bamtobed (v2.26.0). The 47 ARSs, including 8 "confirmed", mapping to different chromosomes in sacCer1 and BT1 assemblies were removed. For the sake of clarity, only the centre of

"confirmed" ARSs are represented in RT profiles throughout this study. BT1 assembly, together with genomic annotations are available on GitHub at https://github.com/LacroixLaurent/NanoTiming.

BT1 genome assembly completeness was estimated by computing its BUSCO (Benchmarking Universal Single-Copy Orthologs) score[60], along with that of S288C R64 (sacCer3) and two W303 long read-based assemblies[61,62] for comparison; BT1 scored as high as S288C "gold standard", validating our assembly (Supplementary Table 2). The quality of BT1 assembly at chromosome extremities was specifically evaluated by assessing chromosome end alignments for each of the aforementioned assemblies. To do so, primary alignments overlapping telomeres (defined by Telofinder[63], https://github.com/GillesFischerSorbonne/telofinder) and/or the 10 kb adjacent window (i.e., subtelomeric regions) were extracted, reads were counted and mapped and soft-clipped (i.e., not part of the alignment) lengths were measured. As indicated in Supplementary Table 2, BT1 recovered at least 1.4 times more reads and aligned sequence length than any other assembly while exhibiting the longest average mapped length and shortest average soft-clipped length, demonstrating its accuracy at chromosome extremities.

### Genomic mean BrdU content profiles

BrdU-labelled reads mapped with megalodon (mod_mappings.bam) were imported into R (v4.0.5) using samtools view (v1.13) and converted into a data frame. For each genomic read, Ml and Mm fields containing BrdU probabilities in an 8-bit integer format and their relative position along the read, respectively, were converted into a probability at each genomic T position, while mitochondrial reads were ignored. BT1 genome was then divided into 1 kb consecutive, non-overlapping bins, and BrdU probabilities along single reads were averaged in each 1 kb bin and used in Fig. 1. To generate genomic mean BrdU content profiles, the mean BrdU content for each 1 kb bin was computed by averaging values from the cognate read-level bins. Only read-level bins with a mean BrdU signal above 0.02, corresponding to the background level of our BrdU basecalling model as determined in ref. 13, were kept in order to remove parental reads and bins replicated before BrdU addition. In parallel, read coverages were computed using bedtools bamtobed (v2.26.0) from mod_mappings.bam files. Bed files were then imported into R using rtracklayer package (v1.62.0) and exported as coverage. In order to compare Nanotiming, sort-seq, and MFA-seq RT profiles while excluding outliers, data values of each experiment were linearly rescaled so that 1 and 2 are the 0.5 and 99.5 percentiles of RT value distribution after rescaling, respectively. Spearman's rank correlation coefficients between RT profiles were computed using the R cor function. Note that rank correlation is blind to the linear rescaling.

In Fig. 1c, for each BrdU labelling concentration, the mean BrdU content value of every 1 kb bin of BT1 genome was plotted against the corresponding sort-seq relative copy number value. Spearman's rank correlation coefficient between both parameters was calculated using the R cor function; a linear regression model fitting the data was also computed. In Supplementary Fig. 1a, the evolution of dTTP level during S phase in BT1 cells was predicted from the mean BrdU content for 5, 10 and 20 μM BrdU labelling doses considering that the mean BrdU content (MBC) mirrors the intracellular BrdUTP (B) to dTTP (T) ratio. Indeed, since MBC = B/(B + T), it follows that T = B*(1/MBC-1). The T term in the equation was computed for every 1 kb bin of the yeast genome from their MBC value, assuming that B equals the labelling BrdU concentration. Finally, to follow dTTP concentration changes in the S phase, the resulting T values were plotted against the sort-seq relative copy number values, normalised between 0 (start of S phase) and 1 (end of S phase), of the cognate bins. An exponential function fitting the data was computed for each BrdU labelling concentration using the R nls function and then implanted into the MBC = B/(B + T) formula to generate a curve that was superimposed to the corresponding experimental distribution of mean BrdU content versus sort-

seq relative copy number (normalised between 0 and 1 as above) data points in Supplementary Fig. 1b.

For Supplementary Fig. 2, nanopore reads of genomic DNA from thymidine-auxotroph MCM869 cells[24] grown for 24 h with various mixtures of BrdU and thymidine, from 0 to 100% BrdU in 10% increments (data from ref. 13), were first mapped to BT1 genome. The mean BrdU content was subsequently computed in 1 kb bins and then plotted versus BT1 sort-seq relative copy number in the cognate bins.

### Evaluation of Nanotiming profiles' quality as a function of sequencing depth

The impact of sequencing depth on Nanotiming profiles' quality was evaluated by calculating, in wild-type BT1 cells, Spearman's rank correlation coefficients between BT1 genome mean BrdU content profiles built from various amounts of nanopore sequencing data and BT1 genome sort-seq profile. Wt_rep1 Nanotiming data were subsampled 100 times, varying either the number of reads between 1000 and 300,000 reads (Supplementary Fig. 22a) or the corresponding genomic coverage between 1x and 600x (Supplementary Fig. 22b). Spearman's rank correlation coefficient was computed between each subsampled Nanotiming profile and BT1 sort-seq profile using R cor function with 'use = "pairwise.complete.obs"' option.

### Estimation of Nanotiming profile intrinsic noise

In order to compute an intrinsic noise estimator for Nanotiming, sort-seq, and MFA-seq profiles, we evaluated the scaled signal difference between 1 kb consecutive bins from the observation that RT signal variation occurs in the scale of five to tens of kilobases, meaning that bin-to-bin variations mostly result from the noise of the tested method. We then used the variance of bin-to-bin signal differences as an estimator of the intrinsic noise. We performed 10 independent subsamplings of Nanotiming, sort-seq and MFA-seq data at various sequencing depths that we estimated with the number of mapped bases (from 1 Mb to 2.5, 5 and 30 Gb for sort-seq, MFA-seq and Nanotiming, respectively). Of note, whereas MFA-seq and sort-seq profiles are established from short sequences (< 300 bp), Nanotiming is based on reads significantly longer than the 1 kb bin size (typical median nanopore read length between 10 and 20 kb, Supplementary Data 4). And because nanopore reads span successive bins, neighbouring bin values are not independent. This dependency reduces noise measurement, favouring Nanotiming over MFA-seq and sort-seq approaches. To remove the dependency between successive bins of nanopore reads, we additionally performed independent random samplings of Nanotiming data at different sequencing depths and compared, for a given number of mapped bases, signal variation between consecutive bins from independent samplings. In order to evaluate the noise for a random profile with no correlation between genomic positions, our noise estimator was also computed after shuffling Nanotiming values between genomic bins. Nanotiming data corresponded to BT1 PromethION dataset with all technical replicates pooled together. MFA-seq data from ref. 8 and our own BT1 sort-seq data were used and processed in accordance with a published pipeline[44] with minor modifications (notably, bins with coverage below a quarter of the median coverage were ignored; see https://github.com/LacroixLaurent/NanoTiming for further details). Results are reported in Supplementary Fig. 23.

### Telomeric repeat length measurement

Raw nanopore signal was basecalled with megalodon (v2.5.0) using the super accuracy model ("dna_r9.4.1_450bps_sup.cfg"). Basecalled reads (.fastq) were processed with Porechop_ABI to remove barcode sequences (https://github.com/bonsai-team/Porechop_ABI; option: "-abi") and were aligned to BT1 assembly using minimap2 (v2.26; option: "-axe map-ont"). Telofinder was used to annotate telomeric regions. For left and right chromosome ends, nucleotide sequences

aligned on the most distal telomeric repeats were extracted, including the eventual soft-clipped bases that may correspond to additional, unmapped telomeric sequences. All candidate sequences were oriented from centromere to telomere. Sequences corresponding to $C_{1-3}A$ telomeric repeats were complemented into $TG_{1-3}$ repeats to simplify downstream processing. Since the inspection of a subset of candidate sequences revealed the presence of non-telomeric repeats, we then filtered candidate sequences based on their nucleotide composition while taking into account the nanopore sequencing basecalling error rate. We defined three criteria for selecting genuine telomeric repeat sequences, which we evaluated through a 50 nucleotide (nt) sliding window: 1. a fraction of GG dinucleotides (freqGG) comprised between 0.05 and 0.4; 2. a sum of the fractions of GG, TG and GT dinucleotides (freqTelo, as in Telofinder) comprised between 0.79 and 1; 3. a Shannon entropy (H) comprised between 0.85 and 1.7 (H is used in Telofinder to filter out complex nanopore sequencing artefacts). In case the telomeric repeat sequence was smaller than 50 nucleotides, the 3 criteria were computed in a single window containing all the nucleotides. Candidate sequences for which at least 90% of the 50 nt windows respected all three criteria were kept to compute the size of telomeric repeats. Please note that TEL13R length was not determined in the *yku70Δ* mutant due to the loss of a Y' element at the right end of chromosome XIII compared to BT1 assembly, preventing proper read mapping and telomeric repeat length measurement; it was therefore removed and excluded from further analyses.

### RT at individual telomeres

Since BT1 genome was divided into 1 kb bins starting from the first nucleotide of each chromosome to compute mean BrdU content profiles, RT at individual telomeres in Fig. 3 was estimated by taking, for a given chromosome end, the average of the rescaled mean BrdU content values (see "Genomic mean BrdU content profiles" section) of the two 1 kb genomic bins of BT1 assembly overlapping the 1 kb adjacent to the terminal telomeric repeats weighted by their relative overlap. A telomere was classified as XY' if the distance between its X element and the closest Y' element on the same chromosome was smaller than 20 kb, otherwise it was classified as an X telomere. This prevented wrong X/XY' assignments in case Y' elements were present on both arms of the same chromosome.

### Comparison between telomere RT and telomere length at the single-molecule level

For each single-molecule mapped on a terminal telomeric repeat, the RT value corresponds to the average BrdU probability in the 1 kb region located immediately next to the terminal telomeric repeats, which is further rescaled based on BrdU signal distribution in the experiment of origin (see "Genomic mean BrdU content profiles" section). The length of the cognate telomeric repeats was determined as described in the "Telomeric repeat length measurement" section. Only reads (i) longer than 5 kb and (ii) with an unscaled single-telomere RT above the background noise of our BrdU detection model (i.e., > 0.02; this filtering exclusively retains BrdU-containing molecules) were conserved. Coefficients and p-values of Spearman's correlation tests (two-sided) between telomere length and RT in the various BT1 strains were computed using the R cor function. Results are presented in Fig. 4b–f and Supplementary Figs. 17–21.

### Sort-seq

Sort-seq was performed on BT1 cells as described in ref. 44 with minor modifications. Cells were grown overnight in YPD, diluted in fresh medium at $OD_{600} \approx 0.1$, harvested after reaching $OD_{600} \approx 0.8$, fixed in 70% ethanol then prepared for fluorescence-activated cell sorting (FACS). Cells were washed with 50 mM sodium citrate pH 7.4, incubated for 1 h at 50 °C in sodium citrate buffer supplemented with

$0.25\,mg.ml^{-1}$ RNAse A, added with $2\,mg.ml^{-1}$ proteinase K and incubated for one additional hour. DNA was counterstained overnight with $1\,\mu M$ SYTOX Green (Invitrogen #S7020). Cells in the S and G2 phases of the cell cycle were sorted using a BD FACSMelody™ Cell Sorter operated by BD FACSChorus v1.3.2 software. DNA was extracted as in ref. 44. Sorting results are presented in Supplementary Fig. 27b. Paired-end sequencing was performed on an Illumina NovaSeq 6000 System. FASTQ files were cleaned with FastXtend (https://www.genoscope.cns.fr/fastxtend/) and processed as in ref. [44] (localMapper, https://github.com/DNAReplicationLab/localMapper/, using bowtie v2.4.4, bedtools v2.30, samtools v1.19, picard v2.7.3; Repliscope, https://github.com/DNAReplicationLab/Repliscope/) with default parameters. The relative copy number was computed in 1 kb windows using the -w 1000 option.

### Reference genome

De novo BT1 genome (see above) was used as the reference genome.

### Computational resources

MFA-seq data from cells of the same genetic background as BT1, namely W303, are from ref. 8 and sort-seq data for *ctf19Δ* and *rif1Δ* cells are from ref. 25. (please note that these data come from homozygous diploid *ctf19Δ* cells and that it has been reported that the RT programme in haploids and diploids is identical[8]) and ref. 7, respectively; relative copy number profiles were generated as described in ref. 44 with default parameters, using BT1 reference genome for mapping. MCM869 nanopore sequencing data are from ref. 13; reads were mapped to BT1 genome.

### Reporting summary

Further information on research design is available in the Nature Portfolio Reporting Summary linked to this article.

## Data availability

Nanopore and Illumina sequencing data generated in this study have been deposited in the ENA database under accession code PRJEB76824. Source data and BT1 assembly with genomic annotations can be found on GitHub (https://github.com/LacroixLaurent/NanoTiming)[64]. S288C R64 (sacCer3) genome assembly (GCF_000146045.2) and W303 assemblies from ref. 61 (GCA_000773925.1) and ref. 62 (GCA_002163515.1) were downloaded from https://www.ncbi.nlm.nih.gov/datasets/genome/GCF_000146045.2/, https://www.ncbi.nlm.nih.gov/datasets/genome/GCA_000773925.1/ and https://www.ncbi.nlm.nih.gov/datasets/genome/GCA_002163515.1/, respectively. MCM869 nanopore sequencing data from ref. 13 are accessible from the ENA repository under accession code PRJEB50302. ARS positions from OriDB[59] are available at http://cerevisiae.oridb.org/. MFA-seq data from ref. 8, sort-seq data for *ctf19Δ* cells from ref. 25 and sort-seq data for *rif1Δ* cells from ref. 7 are accessible from NCBI's Gene Expression Omnibus GEO repository under accession codes GSE48212 (GSM1180746 and GSM1180747 samples), GSE41982 (GSM1029480 and GSM1029481 samples) and GSE97953 (GSM2583609 and GSM2583613 samples), respectively.

## Code availability

Custom scripts used in this study can be found on GitHub (https://github.com/LacroixLaurent/NanoTiming)[64].

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

## Acknowledgements

The authors thank Magali Hennion, Stéphane Marcand, Stefano Mattarocci and all members of the O.H. laboratory for helpful discussions and critical reading of the manuscript, Gilles Fischer for his help in assembling BT1 genome, IBENS GenomiqueENS facility for their assistance with nanopore sequencing, IBENS IT platform and BioClust computing cluster (Labex Memolife) for data management, the Genoscope (Evry, France) for performing Illumina sequencing, and Nathalie Verin for proofreading the manuscript. This work was supported by grants from Fondation pour la Recherche Médicale [FRM EQU202203014910 to O.H.] and Agence Nationale pour la Recherche [NanoPoRep ANR-18-CE45-0002 and HUDROR ANR-19-CE12-0028 to B.A. and O.H.]. B.T. and E.S. were supported by a fellowship from the Ministère de l'Enseignement Supérieur et de la Recherche. B.T. was also supported by a fellowship from the Fondation pour la Recherche Médicale [FRM FDT202106013030].

## Author contributions

B.T. and E.S.C. performed the experiments and the nanopore sequencing. B.T., A.T. and L.L. performed the computational studies. B.T., B.L.T., A.T., L.L., J.M.A., B.A., E.J. and O.H. analysed the data. B.L.T. wrote the manuscript, with inputs from the other authors. B.L.T. and B.T. designed the project. B.L.T. supervised the study.

## Competing interests

The authors declare no competing interests.
