## [Transparent Peer Review file · Nature Communications]

Telomere-to-telomere DNA replication timing profiling using single-molecule sequencing with Nanotiming

Corresponding Author: Dr Benoît Le Tallec

Version 0:

Reviewer comments:

Reviewer #1

(Remarks to the Author)

In this manuscript, Theulot and colleagues describe a new method to measure genome-wide replication timing (RT) profiles using BrdU incorporation and Nanopore sequencing. The method relies on the physiological increase in the dNTP pool during S-phase, which is predicted to result in a competition for incorporation of exogenous BrdU, which becomes increasingly unfavorable over time during replication. By exploiting this property in the budding yeast model, the authors propose a relatively simple method that does not require synchronization or cell sorting and instead only needs growth of an asynchronous culture for the duration of one cell cycle with an adequate concentration of BrdU. Subsequent Nanopore sequencing, base calling and analysis leverage the recent development of many dedicated tools, including several developed by the authors' team.

The RT profiles obtained through this method compare very well with Sort-seq, with the added value of not requiring cell sorting (with the associated caveats) and of being able to analyze repetitive regions including subtelomeres and telomeres. The RT profiles reach a much better resolution compared to MFA-seq, which likewise does not require synchronization or sorting. The authors then used their method to confirm and revisit changes in RT profiles in *yku70Δ*, *rif1Δ*, *ctf19Δ* and *fkh1Δ* mutants, with a particular focus on telomeric regions. They found that in *rif1Δ*, only a subset of chromosome ends are affected in their RT, depending on the composition of the subtelomeres.

Overall, the method presented in this work shows remarkable and well supported results, which will clearly be useful for the community working on replication. The manuscript is well written and the results well presented. However, I do have one major concern and a short list of minor comments.

Major point (divided into several comments and questions):

The method is exclusively dependent on the property of increasing dTTP pool during S phase. Non-trivial reasoning and results (e.g. Fig. 1c) are drawn from this single property and they should be better explained.

How does the dTTP pool increase during S-phase (linear or non-linear)? This should be described in the introduction if previous works already characterized this phenomenon. For instance, a non-linear increase should result in a deformed RT profile compared to Sort-seq, which is not the case.

Lines 88-90, "These results therefore show that S phase dTTP pool fluctuations in S phase can be best interrogated...": the way this sentence is constructed is a shortcut since the results show nothing about dTTP pool fluctuations directly.

Lines 99-101, "in line with our previous conclusion from individual reads that 5 to 20 μ M BrdU concentrations can optimally record the rise in dTTP levels during S phase": again, the authors are not directly measuring dTTP levels. Because it can be difficult to follow, they should better explain the expected correlation between BrdU level and DNA copy number, and how the latter reflects the increase in dTTP levels in S phase. In particular, the linear correlation in Fig. 1c is far from trivial.

Ideally, directly testing if the method indeed relies on an increase in dTTP pool during S-phase could provide a decisive demonstration of how the method works. For example, we would expect that, in contrast to Sort-seq, this method would no longer work in a *sml1Δ* mutant or in strains mutated for RNR genes. Another way to challenge the method would be to test it with the wild-type strain incubated with a low dose of HU, enough to alter (or even completely disrupt) the results based on Nanotiming, but not too high so as not to prevent RT profiling using another method such as Sort-seq. These additional experiments might not be absolutely necessary so long as the authors explain very well the basis of their method in a

convincing way, but they would still be nice to have to reinforce the paper.

By the way, it could be predicted that this method would no longer work in any condition affecting the dNTP pool increase in S-phase, precluding some types of experiments and some mutants. This could be added in the discussion section.

Minor comments:

Title: "nanopore" should be "Nanopore".

Abstract, "by interrogating changes in intracellular dTTP concentration": this sentence is also a shortcut in my opinion and could be rephrased.

Line 83, "our in-house BrdU basecalling model": is it compatible with the latest Nanopore flowcells?

Fig. 1b: can differences between the reads shown be sequence-specific? Eg, higher BrdU content in reads with more Ts.

Lines 94-96, "BrdU-free bins, corresponding either to parental DNA or to DNA replicated before medium supplementation with BrdU, were filtered out": these BrdU-free bins should be quantified and shown, one way or another (most likely in supplementary figures), to allow the reader to evaluate the fraction of reads not analyzed.

Lines 115-116, "Similarly, RT of the repetitive rDNA locus, which was previously inaccessible, could now be retrieved (Fig. 2a, b)": I guess only partially, for the repeats close to the edges of the region, which are the only ones that can be mapped unambiguously.

Line 149 and 274: "In yeasts *S. cerevisiae* and *S. pombe*" should be "In the yeasts *S. cerevisiae* and *S. pombe*".

Line 165: is there an origin in the X element?

Line 174, "telomere shortening in yku70 mutants is held responsible for their early RT": it's not telomere shortening per se, but rather the shorter steady-state of telomere length.

Lines 196-198: what is the fraction of telomeres in yku70 Δ and rif1 Δ mutants that show roughly WT length but still display earlier replication?

Lines 210-224: this part is a bit lengthy. It is interesting to discuss how the method is cost-efficient but some sentences sound like a TV commercial. Maybe the details could be put in the (supplementary) method section? Maybe the authors could also compare the time used to process the samples and how easy or difficult the respective protocols are?

Line 277: this should be explained at the beginning of the results and/or even in the introduction.

Line 286: as well as heterogeneity in telomere length distribution for chromosome-end-specific telomere.

Line 349: the typical doubling time in standard conditions should be indicated.

Line 473: if the size of the sliding window is 50 bp, I am not sure how individual telomere lengths of <50 bp were recovered in the yku70 Δ samples.

Line 370, genome assembly: regarding the chromosome-end-specific analysis of telomere length and RT, it is of crucial importance to have a correct genome assembly. Did the authors compare their assembly with existing long-read-based assemblies of W303 or other syntenic strains? Especially for the regions close to chromosome ends.

Lines 494-497: is there a sequence-specific bias for RT value measurement if the 1 kb-sequence adjacent to the telomere has a particularly low or high level of T/U? Given that subtelomeric sequences often share homology across chromosome ends, this might artificially assign early or late replication status to some groups of extremities.

(Remarks on code availability)

Remarks on code availability: I have checked that the github link works and contains the indicated data and scripts. However, since I am not a bioinformatician, I did not test and review the code.

Reviewer #2

(Remarks to the Author)

Theulot et al., Nanotiming: single-molecule based, telomere-to-telomere DNA replication timing profiling by nanopore sequencing

In this manuscript, the authors developed a novel DNA replication timing (RT) profiling technology called Nanotiming, a single-molecule nanopore sequencing-based telomere-to-telomere (T2T) RT profiling method in budding yeast. Nanotiming measures the percentage of BrdU incorporation in every genomic bin and uses these numbers as a proxy for RT. As dTTP

concentration increases with S-phase progression, BrdU incorporation rate should gradually decrease as S-phase proceeds and can serve as a valuable index for the earliness of RT. Using budding yeast as a model, the authors show that Nanotiming generates genome-wide RT profiles comparable to conventional methods like sort-seq and Repli-seq. Moreover, the power of the T2T reference genome combined with long-read nanopore sequencing allows the authors to profile the RT of previously unapproachable regions of the genome, such as rDNA repeats and telomeres. Genetic mutants involved in telomere/centromere RT regulation (Rif1, Ctf19, yKu70) and other mutants (Fkh1) showed RT defects consistent with what is known in the literature. The authors further discovered that RT control of telomeres by Rif1 is specific to those with X and Y' repetitive elements but not with only X repeats. In addition, the authors could measure the telomere length and RT of individual molecules in wild-type and various mutant cells, confirming and extending the known relationship between them. The genome-wide RT data presented all look beautiful and convincing. I have some comments and suggested experiments (see below), but overall, I recommend this manuscript for publication in Nature Communications.

1. Can the authors obtain similarly high-quality RT profiles when using different BrdU concentrations? The authors only checked a single concentration (5 μ M), but I wonder if similar RT profiles can be obtained if the BrdU concentration is within a specific range. The authors provided data showing the optimal BrdU concentration range for the analyses of BrdU content (Fig. 1b) and its relationship to relative copy number (Fig. 1c). However, the authors did not provide such data regarding the optimal BrdU concentration range for generating RT profiles. Given that S-phase progression is problematic with >10 μ M BrdU (Fig. 1d) and Fig. 2b shows a relatively low BrdU incorporation rate with 5 μ M BrdU, I started to wonder if the optimal BrdU concentration range is minimal and finding the proper range becomes a challenge when applied to a new organism.

2. What is the authors' view regarding the possibility of applying this technology to organisms other than budding yeast? In other words, what is known about the cell-cycle kinetics of dTTP concentration in different organisms? The authors have nicely shown that this methodology works well in budding yeast. However, every aspect of this methodology, including data interpretation, seems to depend on dTTP kinetics during cell cycle progression. While relevant references are cited (although most are yeast studies), and the authors do talk about the importance of BrdU concentration in the Discussion section, I feel that the details of this critical factor are not provided or discussed thoroughly in this manuscript.

3. Lines 143-146. It reads as if the Nanotiming protocol per se was the key to achieving T2T RT profiling. However, if you simply combine conventional Repli-seq, long-read sequencing, and T2T reference genome, it seems that you can achieve T2T RT profiling. I do agree that measuring the BrdU incorporation rate is a clever idea, and it works beautifully in budding yeast. However, I feel that the description here needs to be a bit more reserved to avoid misunderstanding because the combination of long-read sequencing and the T2T reference genome is really the key to achieving T2T RT profiling.

4. The cost of conventional RT profiling of the 12-Mb genome of the budding yeast is set way too high, given the tiny genome size. Conventional Repli-seq profiling of human/mouse cells should only cost <200 USD. The authors should re-do the calculation. Alternatively, the authors could tone down on the cost discussion in the first place. I understand that cost is important for doing science. However, it seems rather inappropriate in a research paper, especially in the Discussion. The authors should focus more on the scientific strength of their novel method.

Other points:

- Line 93: The sort-seq principle should be briefly described in the main text (i.e., S and G2 sort followed by NGS and taking the ratio) – it would be helpful for the readers.
- Figures 2,3,4, and all other relevant Supplementary Figures: the 'RT' label is difficult to understand for non-yeast biologists. Labeling 'early' and 'late' on the Y-axes would help.
- Figure 2: rDNA cluster position should be labeled in the figure
- Figure 3: (typo) The genomic coordinates are probably not in kb (kilobases).
- Figure 4: it might have been less confusing if the authors used the same X-axis scale for Figures 4b, c, d, e, and f.

(Remarks on code availability)

Reviewer #3

(Remarks to the Author)

Theulot et al., introduce a novel method, Nanotiming, for profiling DNA replication timing (RT) using nanopore sequencing. The approach is novel and addresses some of the limitations of other approaches. The ability to assay unsynchronised and unsorted cells is a major advantage as is the measurement replication dynamics across the entire genome is a notable improvement although one that might be achieved if other approaches also exploited long reads. The paper is well written and presented and the data clearly supports the authors claims that Nanotiming is a powerful and robust approach and I found little that required further expansion. A nice extension of this study would be to apply it to more complex genomes such as metazoans, but I appreciate that this is beyond the scope of the current manuscript.

Major comment:

-The authors demonstrate the cost effectiveness of the Nanotiming by subsampling their sequencing coverage to 1Gb. However the same approach is not taken for alternative methods such as sort-seq. Can the authors please demonstrate that similar cost-effectiveness is not seen with lower coverage in other techniques.

(Remarks on code availability)

Version 1:

Reviewer comments:

Reviewer #1

(Remarks to the Author)

My initial assessment of this manuscript was already positive. The authors have now addressed the concerns I had with additional analyses and experiments, in a satisfactory way. I thus support the publication of the manuscript.

(Remarks on code availability)

Although I checked the presence of the codes and the recent modifications, I do not have the expertise to review the codes.

Reviewer #2

(Remarks to the Author)

Within a relatively short time, the authors have fully addressed all the points I raised and related points raised by Reviewer #1. I congratulate the authors for their excellent work.

(Remarks on code availability)

Reviewer #3

(Remarks to the Author)

The authors have included a more balanced discussion of the advantages of Nanotiming and decreased the emphasis on cost as an advantage. I am happy to support the publication of this excellent manuscript.

(Remarks on code availability)

Response to referees

- **Reviewer #1**

In this manuscript, Theulot and colleagues describe a new method to measure genome-wide replication timing (RT) profiles using BrdU incorporation and Nanopore sequencing. The method relies on the physiological increase in the dNTP pool during S-phase, which is predicted to result in a competition for incorporation of exogenous BrdU, which becomes increasingly unfavorable over time during replication. By exploiting this property in the budding yeast model, the authors propose a relatively simple method that does not require synchronization or cell sorting and instead only needs growth of an asynchronous culture for the duration of one cell cycle with an adequate concentration of BrdU. Subsequent Nanopore sequencing, base calling and analysis leverage the recent development of many dedicated tools, including several developed by the authors' team. The RT profiles obtained through this method compare very well with Sort-seq, with the added value of not requiring cell sorting (with the associated caveats) and of being able to analyze repetitive regions including subtelomeres and telomeres. The RT profiles reach a much better resolution compared to MFA-seq, which likewise does not require synchronization or sorting. The authors then used their method to confirm and revisit changes in RT profiles in *yku70Δ*, *rif1Δ*, *ctf19Δ* and *fkh1Δ* mutants, with a particular focus on telomeric regions. They found that in *rif1Δ*, only a subset of chromosome ends are affected in their RT, depending on the composition of the subtelomeres. Overall, the method presented in this work shows remarkable and well supported results, which will clearly be useful for the community working on replication. The manuscript is well written and the results well presented.

We thank Reviewer #1 for these comments.

However, I do have one major concern and a short list of minor comments.

Major point (divided into several comments and questions):

- The method is exclusively dependent on the property of increasing dTTP pool during S phase. Non-trivial reasoning and results (e.g. Fig. 1c) are drawn from this single property and they should be better explained.
- How does the dTTP pool increase during S-phase (linear or non-linear)? This should be described in the introduction if previous works already characterized this phenomenon. For instance, a non-linear increase should result in a deformed RT profile compared to Sort-seq, which is not the case.

To better understand the relationship between mean BrdU content and time, we extracted from 5, 10 and 20 μM BrdU data points in Fig. 1c the dynamics of dTTP concentration during S phase. Indeed, considering that the mean BrdU content (MBC) reflects the intracellular BrdUTP (B) to dTTP (T) ratio ($\text{MBC} = \text{B}/(\text{B} + \text{T})$), dTTP pool can be estimated by inverting the terms of the equation ($\text{T} = \text{B} * (1/\text{MBC} - 1)$). Satisfactorily, all three concentrations recovered the exponentially-shaped dTTP level increase in S phase seen in *S. cerevisiae* (Koc et al, 2003) (new Supplementary Fig. 1a). To further illustrate the mathematical relationship connecting dTTP level, mean BrdU content and time, the exponential function that best fitted the increase in dTTP concentration in S was in turn inserted in the $\text{MBC} = \text{B}/(\text{B} + \text{T})$ formula, recapitulating the observed, quasi-linear mean BrdU content decrease in the same time interval (new Supplementary Fig. 1b). These results, which are now included in the manuscript (lines 110-119), therefore strongly suggest that it is actually the exponential growth of the dTTP pool during S phase that causes the mean BrdU content to decrease in a near-linear fashion in the

meantime. Please also see below for our direct demonstration that Nanotiming indeed relies on S phase dTTP pool increase.

Supplementary Figure 1. Relationship between dTTP level and mean BrdU content in S phase. **a**, Predicted evolution of dTTP level during S phase in BT1 cells. For 5, 10 and 20 µM BrdU labelling doses, the mean BrdU content value (MBC) of every 1 kb bin of BT1 genome (y coordinate of Fig. 1c data points) was converted into a dTTP concentration (T) based on the formula $T=B*(1/MBC-1)$, assuming that B equals the BrdU labelling concentration. The resulting T values were then plotted against the corresponding sort-seq relative copy number values (x coordinate of Fig. 1c data points) normalised between 0 and 1 corresponding to the start and end of S phase, respectively; this amounts to following dTTP level in the course of S phase. All three BrdU labelling concentrations recovered a similar, exponentially-shaped increase of dTTP during S phase. Coloured curves, exponential fits of the data. **b**, Mean BrdU content versus sort-seq relative copy number in 1 kb bins of BT1 genome as in Fig. 1c except that sort-seq data were normalised between 0 (start of S phase) and 1 (end of S phase) as in **a**. For each BrdU labelling concentration, the coloured curve was computed by implanting the exponential function determined in **a** into the $MBC=B/(B+T)$ formula. **a**, **b**, See text and Methods for details.

The following paragraph has also been incorporated in the Discussion section (lines 285-296): “In addition to yeast (Koc et al, 2003; Chabes et al, 2003; Koc et al, 2004; Hakansson et al, 2006), several studies have documented that intracellular dTTP levels expand in the course of S phase in mammalian cells (e.g., Nordenskjold et al, 1970; Walters et al, 1973; McCormick et al, 1983; Kenigsberg et al, 2016). It is thus tempting to speculate that dTTP increase during S phase is a feature shared by all eukaryotes, especially considering that nucleotide biosynthesis pathways are evolutionarily conserved. Still, it is impossible to guarantee that dTTP pool kinetics throughout S phase in every eukaryotic cell will resemble the exponential increase in

S. cerevisiae which results in a near-linear relationship between mean BrdU content and replication time. It should however be noted that even in the case of a non-linear relationship between both parameters, accessing RT remains possible as long as this relationship is strictly monotonic (i.e., if the mean BrdU content is always decreasing over time, regardless of how it decreases) since a strictly monotonic function can be inverted mathematically. We are therefore confident that Nanotiming can be applied to a broad range of eukaryotic organisms”.

- Lines 88-90, “These results therefore show that S phase dTTP pool fluctuations in S phase can be best interrogated...”: the way this sentence is constructed is a shortcut since the results show nothing about dTTP pool fluctuations directly.

We hope that our additional explanations and results (see above and below) will have convinced Reviewer #1 that Nanotiming indeed interrogates dTTP pool fluctuations in S phase to profile replication timing; “best interrogated” has also been replaced by “indirectly interrogated” (line 92).

- Lines 99-101, “in line with our previous conclusion from individual reads that 5 to 20 μ M BrdU concentrations can optimally record the rise in dTTP levels during S phase”: again, the authors are not directly measuring dTTP levels. Because it can be difficult to follow, they should better explain the expected correlation between BrdU level and DNA copy number, and how the latter reflects the increase in dTTP levels in S phase. In particular, the linear correlation in Fig. 1c is far from trivial.

Please see our additional explanations and results above and below; “record” has also been replaced by “catch” (line 106).

- Ideally, directly testing if the method indeed relies on an increase in dTTP pool during S-phase could provide a decisive demonstration of how the method works. For example, we would expect that, in contrast to Sort-seq, this method would no longer work in a *sml1 Δ* mutant or in strains mutated for RNR genes. Another way to challenge the method would be to test it with the wild-type strain incubated with a low dose of HU, enough to alter (or even completely disrupt) the results based on Nanotiming, but not too high so as not to prevent RT profiling using another method such as Sort-seq. These additional experiments might not be absolutely necessary so long as the authors explain very well the basis of their method in a convincing way, but they would still be nice to have to reinforce the paper.

To directly demonstrate that Nanotiming indeed relies on an increase in dTTP pool during S phase, we thought of using the MCM869 strain, inactivated for the thymidylate synthase-encoding *CDC21* gene required for *de novo* dTMP biosynthesis and therefore completely deprived of endogenous dTTPs; to grow, this strain relies on exogenous thymidine or its analogues, which are phosphorylated into dTMP usable by the cell thanks to an integrated Herpes simplex virus thymidine kinase (hsvTK) transgene under the control of the constitutive GPD promoter (Ma et al, 2012). Because of their lack of dTTP level increase in S phase, BrdU-labelled MCM869 yeasts are predicted to yield nanopore reads with homogenous BrdU levels. Accordingly, the mean BrdU content computed from reads of genomic DNA of MCM869 cells grown with various mixtures of BrdU and thymidine (from 0 to 100 % BrdU) in the culture medium remained constant throughout S phase, with a value that simply echoed the extracellular proportion of BrdU (new Supplementary Fig. 2). These results have been included in the manuscript (lines 119-132).

Supplementary Figure 2. Comparison between relative copy number established by sort-seq in 1 kb bins of BT1 genome and mean BrdU content in the cognate bins computed from nanopore reads of genomic DNA of MCM869 cells grown with different proportions of BrdU in the culture medium. BrdU percentages range from 0 (thymidine control) to 100% in 10% increments; MCM869 data are from Theulot et al, 2022; BT1 sort-seq data were used as a proxy for MCM869's since both strains share the same replication program (Theulot et al, 2022).

- By the way, it could be predicted that this method would no longer work in any condition affecting the dNTP pool increase in S-phase, precluding some types of experiments and some mutants. This could be added in the discussion section.

Reviewer#1 is correct. The sentence “By its very nature (...) this method will not work under conditions preventing dTTP pool change in S phase, whether due to mutations in dNTP metabolism genes or the use of certain drugs” has been added in the Discussion section (lines 296-298).

Minor comments:

- Title: “nanopore” should be “Nanopore”.

“Nanopore” is usually written without a capital "N".

- Abstract, “by interrogating changes in intracellular dTTP concentration”: this sentence is also a shortcut in my opinion and could be rephrased.

We hope Reviewer #1 will now be convinced that Nanotiming does interrogate changes in intracellular dTTP concentration in S phase to profile replication timing.

- Line 83, “our in-house BrdU basecalling model”: is it compatible with the latest Nanopore flowcells?

Our in-house BrdU basecalling model is not yet compatible with Oxford Nanopore Technologies' latest R10 chemistry. However, as illustrated in Figure R1 below, we have obtained preliminary results indicating that quality RT profiles can be computed from Nanotiming experiments sequenced on R10.4.1 flow cells using DNAscent 4.0.3 BrdU basecaller from Michael Boemo's laboratory (<https://github.com/MBoemo/DNAscent>).

Figure R1. Nanotiming profiles of *S. cerevisiae* chromosomes using Oxford Nanopore Technologies' R9.4.1 or R10.4.1 flow cells. Mean BrdU content was computed in 1 kb bins using R9.4.1 or R10.4.1 nanopore reads of the same genomic DNA coming from BT1 cells labelled with 5 μ M BrdU for one doubling time (R9.4.1 and R10.4.1 reads were basecalled with our in-house BrdU model or with DNAscent 4.0.3, respectively); data were rescaled between 1 (end of S phase) and 2 (start of S phase). Purple and green vertical lines, positions of confirmed

S. cerevisiae replication origins and of centromeres, respectively; grey box, rDNA; RT, replication timing. R9 and R10 RT profiles are largely superimposable, although BrdU detection within rDNA repeats seems problematical with DNAscent 4.0.3.

- **Fig. 1b: can differences between the reads shown be sequence-specific? Eg, higher BrdU content in reads with more Ts.**

Our BrdU basecaller estimates the probability of having a BrdU at each thymidine site, with the BrdU content of a given region corresponding to the fraction of the thymidine sites that incorporated a BrdU. In other words, the BrdU content is equivalent to the B/(B+T) ratio, which should be independent of the amount of Ts. To ascertain this, we compared the mean BrdU content (mean B/(B+T) ratio) in 1 kb bins of BT1 genome with the AT content in the cognate bins (Figure R2) and found no correlation between both parameters.

Of note, what the BrdU content precisely corresponds to is now defined in the manuscript (see Results section lines 85-86, “Strong variations in BrdU content (i.e., in the proportion of BrdU-substituted thymidine sites)” and Fig. 1b caption, “The BrdU content in 1 kb bins along individual nanopore reads, corresponding to the fraction of the thymidine sites of these bins that incorporated a BrdU (i.e., to their BrdU/(BrdU+Thymidine) ratio), is represented as a 1D heatmap”).

Figure R2. Comparison between mean BrdU content (B/(B+T) ratio) and AT content in 1 kb bins of BT1 genome. Mean BrdU content was computed from nanopore reads of genomic DNA of cells labelled with 5 μ M BrdU for one doubling time.

- **Lines 94-96, “BrdU-free bins, corresponding either to parental DNA or to DNA replicated before medium supplementation with BrdU, were filtered out”: these BrdU-free bins should be quantified and shown, one way or another (most likely in supplementary figures), to allow the reader to evaluate the fraction of reads not analyzed.**

We calculated that 55.6, 51.1 and 52.2% of the bins were filtered out for the 5, 10 and 20 μ M samples, respectively, close to the expected 50% fraction corresponding to parental DNA given that cells had been labelled with BrdU for one doubling time. The observed value slightly over 50% is explained by the presence of BrdU-free regions within nanopore reads corresponding to DNA replicated prior to medium supplementation with BrdU. The corresponding sentence in the Results section has been modified as follows (lines 99-101): “BrdU-free bins of nanopore reads, representing about half of the data (55.6, 51.1 and 52.2% for 5, 10 and 20 μ M, respectively) and corresponding either to parental DNA or to DNA replicated before medium supplementation with BrdU, were filtered out”.

- Lines 115-116, “Similarly, RT of the repetitive rDNA locus, which was previously inaccessible, could now be retrieved (Fig. 2a, b)”: I guess only partially, for the repeats close to the edges of the region, which are the only ones that can be mapped unambiguously.

Although reads with “external” rDNA repeats flanked by sequences located upstream or downstream of the rDNA locus allowing unambiguous mapping contribute to the BrdU signal, most of the reads aligned to the rDNA locus only contain repeats and are therefore randomly mapped on the 12 rDNA copies of BT1 assembly. This gives rise to an average RT profile of the rDNA locus as a whole. This has been indicated in Figure 2 caption.

- Line 149 and 274: “In yeasts *S. cerevisiae* and *S. pombe*” should be “In the yeasts *S. cerevisiae* and *S. pombe*”.

The sentence has been modified accordingly (line 178).

- Line 165: is there an origin in the X element?

All X elements comprise an origin, as indicated in lines 184-186 (“Since *S. cerevisiae* chromosome extremities have distinct subtelomeric regions comprising a single X and none to multiple Y’ repetitive elements, each containing a replication origin (Wellinger and Zakian, 2012)”). It should be noted that not all X nor Y’ origins are present in databases, most likely due to the repetitive nature of X and Y’ elements preventing unambiguous origin attribution, which explains why they seldom appear as purple vertical lines in our RT profiles.

- Line 174, “telomere shortening in *yku70* mutants is held responsible for their early RT”: it’s not telomere shortening per se, but rather the shorter steady-state of telomere length.

The sentence has been modified as follows: “the short telomere length in *yku70* mutants is held responsible for their early RT” (line 203).

- Lines 196-198: what is the fraction of telomeres in *yku70* Δ and *rif1* Δ mutants that show roughly WT length but still display earlier replication?

As illustrated in Figure R3, we calculated that 5.12 and 8.89 % of telomeres in *rif1* Δ and *yku70* Δ mutants, respectively, have wild-type length but display earlier replication.

Figure R3. Length and RT of individual telomeres in *rif1* Δ and *yku70* Δ cells. RT versus length of single telomeres is plotted as a 2D density plot in the strain indicated above each panel. Vertical dotted lines delineate the 95% interval of wild-type telomere lengths; the horizontal dotted line at RT value of 1.5 separates early (top) from late (bottom) replicating telomeres. The percentage of telomeres in a given section of a panel is indicated.

- Lines 210-224: this part is a bit lengthy. It is interesting to discuss how the method is cost-efficient but some sentences sound like a TV commercial. Maybe the details could be put in the (supplementary) method section? Maybe the authors could also compare the time used to process the samples and how easy or difficult the respective protocols are?

We have shortened this part of the manuscript, especially by removing the comparison with the cost of a sort-seq profile, which is difficult to estimate as pointed out by Reviewer#2. Regarding time and protocols of the respective methods, it is now written in the Discussion section (lines 280-284) that “Lastly, Nanotiming takes less than a week (Fig. 1a) and can be fully performed in-house, as it only demands an affordable ONT device. In comparison, although the sort-seq procedure theoretically takes about the same amount of time as Nanotiming (Batrakou et al, 2020), the cell sorting and NGS sequencing it requires are generally only accessible from facilities or companies, which often means a longer processing time” (please note that the reference to Fig. 1a, which displays Nanotiming’s usual timeline, has been added here). As for experimental procedures *per se*, Nanotiming’s elimination of cell sorting is undoubtedly a strong asset in terms of simplicity, as already mentioned throughout the manuscript.

- Line 277: this should be explained at the beginning of the results and/or even in the introduction.

The BT1 strain is now described in more detail at the beginning of the Results section, lines 72-76: “Since fungi naturally lack a thymidine salvage pathway needed for the uptake of extracellular thymidine and its analogues, we used the BT1 strain, which is amenable to highly efficient BrdU incorporation thanks to the joint expression of human equilibrative nucleoside transporter 1 (hENT1) and Herpes simplex virus thymidine kinase (hsvTK) converting BrdU into cell-usable BrdU monophosphate (Theulot et al, 2022)”.

- Line 286: as well as heterogeneity in telomere length distribution for chromosome-end-specific telomere.

The sentence has been modified as follows: “Telomere length heterogeneity at a given chromosome end together with interchromosomal variability in the number of telomeric repeats and in subtelomeric regions make each telomere unique” (lines 329-331).

- Line 349: the typical doubling time in standard conditions should be indicated.

The typical doubling time (90 min) is now indicated in the “BrdU labelling” section of the Methods (line 394).

- Line 473: if the size of the sliding window is 50 bp, I am not sure how individual telomere lengths of <50 bp were recovered in the yku70Δ samples.

In case the telomeric repeat sequence was smaller than 50 nucleotides, the 3 criteria were computed in a single window containing all the nucleotides. This is now indicated in the “Telomeric repeat length measurement” section of the Methods (lines 558-560).

- Line 370, genome assembly: regarding the chromosome-end-specific analysis of telomere length and RT, it is of crucial importance to have a correct genome assembly. Did the authors compare their assembly with existing long-read-based assemblies of W303 or other syntenic strains? Especially for the regions close to chromosome ends.

BT1 genome assembly completeness was estimated by computing its BUSCO (Benchmarking Universal Single-Copy Orthologs) score (Simão et al, 2015), along with that of S288C R64 (sacCer3) and two W303 long read-based assemblies (Berlin et al, 2015; Matheson et al, 2017) for comparison; BT1 scored as high as S288C “gold standard”, validating our assembly (see

new Supplementary Table 6). The quality of BT1 assembly at telomeres was specifically assessed by characterising chromosome end alignments for each of the aforementioned assemblies. As indicated in new Supplementary Table 6, which reports the number of primary mapped reads as well as measured mapped and soft-clipped (i.e., not part of the alignment) lengths, BT1 recovered at least 1.4 times more reads and aligned sequence length than any other assembly while exhibiting the longest average mapped length and shortest average soft-clipped length, demonstrating its accuracy at chromosome extremities.

The above paragraph is now part of “BT1 genome assembly” section of the Methods (lines 453-464).

- Lines 494-497: is there a sequence-specific bias for RT value measurement if the 1 kb-sequence adjacent to the telomere has a particularly low or high level of T/U? Given that subtelomeric sequences often share homology across chromosome ends, this might artificially assign early or late replication status to some groups of extremities.

As shown above, the BrdU content of a 1 kb window reflects its B/(B+T) ratio, which is blind to the number of T sites. Moreover, if the AT content of a given telomere-adjacent 1 kb bin was dictating its BrdU content value, then all yeast strains should systematically exhibit a similar value in that bin, whereas we observe that it greatly differs depending on the genetic context (for instance, the very same 1 kb bin adjacent to TEL01R exhibits a low BrdU content in wild-type cells but a high BrdU content in *rif1*Δ mutant, as shown in Fig. 3a).

- **Reviewer #2**

In this manuscript, the authors developed a novel DNA replication timing (RT) profiling technology called Nanotiming, a single-molecule nanopore sequencing-based telomere-to-telomere (T2T) RT profiling method in budding yeast. Nanotiming measures the percentage of BrdU incorporation in every genomic bin and uses these numbers as a proxy for RT. As dTTP concentration increases with S-phase progression, BrdU incorporation rate should gradually decrease as S-phase proceeds and can serve as a valuable index for the earliness of RT. Using budding yeast as a model, the authors show that Nanotiming generates genome-wide RT profiles comparable to conventional methods like sort-seq and Repli-seq. Moreover, the power of the T2T reference genome combined with long-read nanopore sequencing allows the authors to profile the RT of previously unapproachable regions of the genome, such as rDNA repeats and telomeres. Genetic mutants involved in telomere/centromere RT regulation (Rif1, Ctf19, yKu70) and other mutants (Fkh1) showed RT defects consistent with what is known in the literature. The authors further discovered that RT control of telomeres by Rif1 is specific to those with X and Y' repetitive elements but not with only X repeats. In addition, the authors could measure the telomere length and RT of individual molecules in wild-type and various mutant cells, confirming and extending the known relationship between them. The genome-wide RT data presented all look beautiful and convincing. I have some comments and suggested experiments (see below), but overall, I recommend this manuscript for publication in Nature Communications.

We thank Reviewer #2 for these comments.

1. Can the authors obtain similarly high-quality RT profiles when using different BrdU concentrations? The authors only checked a single concentration (5 μM), but I wonder if similar RT profiles can be obtained if the BrdU concentration is within a specific range. The authors provided data showing the optimal BrdU concentration range for the analyses of BrdU content (Fig. 1b) and its relationship to relative copy number (Fig. 1c). However, the authors did not provide such data regarding the optimal BrdU concentration range for generating RT profiles. Given that S-phase progression is problematic with $>10 \mu\text{M}$ BrdU (Fig. 1d) and Fig. 2b shows a relatively low BrdU incorporation rate with 5 μM BrdU, I started to wonder if the optimal BrdU concentration range is minimal and finding the proper range becomes a challenge when applied to a new organism.

Even if cell labelling for one doubling time with 10 μM BrdU slightly alters S phase progression, the resulting RT profile is indistinguishable from the one obtained from cells treated with 5 μM BrdU, as well as from the sort-seq profile; cell labelling with 20 μM BrdU also gives rise to a very similar RT profile, albeit with somewhat shallower valleys. In conclusion, Nanotiming experiments performed with BrdU concentrations in the 5-20 μM range generate high-quality RT profiles. This is now indicated in the Discussion section (lines 314-316) and illustrated in new Supplementary Figure 26.

Supplementary Figure 26. Mean BrdU content profiles of *S. cerevisiae* chromosomes computed from reads of genomic DNA of BT1 cells labelled with 5, 10 or 20 μM BrdU. Mean BrdU content was calculated in 1 kb bins using nanopore reads of genomic DNA from BT1 cells labelled for one doubling time with 5, 10 or 20 μM BrdU; BT1 sort-seq relative copy number profile computed in 1 kb bins is also shown; data were rescaled between 1 (end of S phase) and 2 (start of S phase) for comparison. Purple and green vertical lines, positions of confirmed *S. cerevisiae* replication origins and of centromeres, respectively; grey box, rDNA; RT, replication timing.

2. What is the authors' view regarding the possibility of applying this technology to organisms other than budding yeast? In other words, what is known about the cell-cycle kinetics of dTTP concentration in different organisms? The authors have nicely shown that this methodology

works well in budding yeast. However, every aspect of this methodology, including data interpretation, seems to depend on dTTP kinetics during cell cycle progression. While relevant references are cited (although most are yeast studies), and the authors do talk about the importance of BrdU concentration in the Discussion section, I feel that the details of this critical factor are not provided or discussed thoroughly in this manuscript.

The following paragraph has been added to the Discussion section (lines 285-296): “In addition to yeast (Koc et al, 2003; Chabes et al, 2003; Koc et al, 2004; Hakansson et al, 2006), several studies have documented that intracellular dTTP levels expand in the course of S phase in mammalian cells (e.g., Nordenskjold et al, 1970; Walters et al, 1973; McCormick et al, 1983; Kenigsberg et al, 2016). It is thus tempting to speculate that dTTP increase during S phase is a feature shared by all eukaryotes, especially considering that nucleotide biosynthesis pathways are evolutionarily conserved. Still, it is impossible to guarantee that dTTP pool kinetics throughout S phase in every eukaryotic cell will resemble the exponential increase in *S. cerevisiae* which results in a near-linear relationship between mean BrdU content and replication time. It should however be noted that even in the case of a non-linear relationship between both parameters, accessing RT remains possible as long as this relationship is strictly monotonic (i.e., if the mean BrdU content is always decreasing over time, regardless of how it decreases) since a strictly monotonic function can be inverted mathematically. We are therefore confident that Nanotiming can be applied to a broad range of eukaryotic organisms”.

3. Lines 143-146. It reads as if the Nanotiming protocol per se was the key to achieving T2T RT profiling. However, if you simply combine conventional Repli-seq, long-read sequencing, and T2T reference genome, it seems that you can achieve T2T RT profiling. I do agree that measuring the BrdU incorporation rate is a clever idea, and it works beautifully in budding yeast. However, I feel that the description here needs to be a bit more reserved to avoid misunderstanding because the combination of long-read sequencing and the T2T reference genome is really the key to achieving T2T RT profiling.

Long-read sequencing together with the T2T reference genome, not Nanotiming *per se*, are indeed instrumental for T2T RT profiling. The “telomere-to-telomere” part of the sentence has therefore been removed. The sentence now reads as follows: “In conclusion, our study demonstrates that it is possible to build high-resolution RT profiles of a eukaryotic genome by measuring changes in intracellular dTTP concentration during S phase through competition with BrdUTP for incorporation into DNA. We named this approach « Nanotiming »” (lines 172-175).

4. The cost of conventional RT profiling of the 12-Mb genome of the budding yeast is set way too high, given the tiny genome size. Conventional Repli-seq profiling of human/mouse cells should only cost <200 USD. The authors should re-do the calculation. Alternatively, the authors could tone down on the cost discussion in the first place. I understand that cost is important for doing science. However, it seems rather inappropriate in a research paper, especially in the Discussion. The authors should focus more on the scientific strength of their novel method.

We indicated in the main text the price we actually paid for the sort-seq profile of the BT1 genome, but we agree with Reviewer#2 that NGS costs can vary widely. We have therefore toned down on the cost discussion in the manuscript by (i) removing “for one-tenth of the cost” from the abstract; (ii) replacing “Current RT mapping techniques also remain expensive. For instance, the cost of an RT profile by sort-seq of the 12 Mb-genome of *Saccharomyces cerevisiae* model eukaryote is about US\$1,000 considering flow cytometry and next-generation sequencing fees. Cost issues are even more critical for metazoan genomes, the large size of which magnifies sequencing expenses” by “Current RT mapping techniques also remain

expensive when adding up the costs of flow cytometry and next-generation sequencing” (lines 45-46); (iii) removing the sentence “By way of comparison, BT1 genomic sort-seq profile prepared according to published guidelines (Batrakou et al, 2020) costed us around US\$1,000” in the Results section.

Other points:

- Line 93: The sort-seq principle should be briefly described in the main text (i.e., S and G2 sort followed by NGS and taking the ratio) – it would be helpful for the readers.

The sort-seq principle is now described in the Results section (“the relative copy number obtained by sort-seq (i.e., the ratio between the number of copies of a given locus in replicating and non-replicating cells estimated through NGS sequencing of genomic DNA from S- and G1/G2-sorted cells”), lines 96-98).

- Figures 2,3,4, and all other relevant Supplementary Figures: the 'RT' label is difficult to understand for non-yeast biologists. Labeling 'early' and 'late' on the Y-axes would help.

'Early' and 'late' labels have been inserted in Fig. 2a as an example for all similar figures.

- Figure 2: rDNA cluster position should be labeled in the figure.

The rDNA locus is now labelled in all RT figures.

- Figure 3: (typo) The genomic coordinates are probably not in kb (kilobases).

We thank Reviewer#2 for spotting the typo, which has been corrected.

- Figure 4: it might have been less confusing if the authors used the same X-axis scale for Figures 4b, c, d, e, and f.

The same X-axis scale is now used for Figures 4b-f.

Reviewer #3

Theulot et al., introduce a novel method, Nanotiming, for profiling DNA replication timing (RT) using nanopore sequencing. The approach is novel and addresses some of the limitations of other approaches. The ability to assay unsynchronised and unsorted cells is a major advantage as is the measurement replication dynamics across the entire genome is a notable improvement although one that might be achieved if other approaches also exploited long reads. The paper is well written and presented and the data clearly supports the authors claims that Nanotiming is a powerful and robust approach and I found little that required further expansion. A nice extension of this study would be to apply it to more complex genomes such as metazoans, but I appreciate that this is beyond the scope of the current manuscript.

We thank Reviewer #3 for these comments. Regarding applying Nanotiming to more complex genomes such as metazoans, please see our answer to Reviewer#2 and the dedicated text in the Discussion section (lines 285-296).

Major comment:

The authors demonstrate the cost effectiveness of the Nanotiming by subsampling their sequencing coverage to 1Gb. However the same approach is not taken for alternative methods such as sort-seq. Can the authors please demonstrate that similar cost-effectiveness is not seen with lower coverage in other techniques.

In addition to Nanotiming, we have now applied our subsampling procedure to sort-seq and MFA-seq data (see new Supplementary Fig. 23, former Supplementary Fig. 21, below), showing that Nanotiming and sort-seq profiles at 1 kb resolution generated from 1 Gb of mapped sequences display an equivalent, low level of noise indicative of a quality RT profile, whereas that of MFA-seq is three times higher. Please note that in response to comments from Reviewers #1 and #2, we have toned down on the cost discussion in the manuscript, and that we do not compare the cost efficiency of Nanotiming and alternative methods any more. The sentence “This indicates that low amounts of sequencing data are sufficient to generate reliable RT profiles by Nanotiming, probably because of an inferior intrinsic noise compared to sort-seq and MFA-seq methods (Supplementary Fig. 21)” is now “This indicates that small amounts of sequencing data are sufficient to produce reliable RT profiles by Nanotiming, probably because of a low intrinsic noise (Supplementary Fig. 23)”, lines 236-238.

Supplementary Figure 23. Evaluation of Nanotiming, sort-seq and MFA-seq noise as a function of the number of mapped bases. **a**, Noise estimator corresponds to the variance of signal differences between consecutive 1 kb bins. Reads were randomly selected from the complete BT1 PromethION or sort-seq datasets or from MFA-seq data from Muller et al, 2014; subsampling was performed 10 times for each category of number of mapped bases. Since nanopore reads are typically 10 to 20 kb in length and therefore span successive 1 kb bins, which makes neighbouring bin values not independent and may artificially reduce noise measurement, signal variation for Nanotiming was compared either between consecutive bins from the same sampling, which preserves the bin-to-bin dependency in nanopore reads (Nanotiming curve, in red) or from independent samplings, which removes this dependency (Nanotiming-uncor curve, in orange). Noise in a random RT profile was estimated using shuffled Nanotiming data (Nanotiming-shuffled, in pink) for comparison. See Methods. **b**, Percentage of *S. cerevisiae* genome covered by subsampled reads for each category of number of mapped bases. **a**, **b**, Horizontal line, median; boxes, 25th to 75th percentiles; whiskers, 1.5x interquartile range. Uncor, uncorrelated.

List of additional changes

Title : “single-molecule based” has been corrected to “single-molecule-based”.

Benjamin Audit’s affiliation has been modified to meet his institution's instructions.

Jean-Michel Arbona’s present address has been added.

Line 28: the abstract has been slightly modified.

Line 51: three additional references (Nordenskjold et al, 1970; Walters et al, 1973 and McCormick et al, 1983) have been included.

Line 89: "content" has been replaced by "level".

Line 206: an additional reference (Schmidt et al, 2024) has been included.

Line 236: "sequenced" has been replaced by "mapped".

Line 261: "Accordingly" has been replaced by "In line with our results”.

Line 279: “generated” has been replaced by “built”.

Line 281: “one week” has been replaced by “a week”.

Lines 317-320: sentences have been slightly modified to avoid redundancy in presentation of the thymidine salvage pathway and of the BT1 strain, which are now described at the beginning of the Results section.

Line 332: “ends” has been replaced by “extremities”.

Lines 389-390: it was previously mentioned that pBL vectors had been deposited on Addgene repository. Unfortunately, CNRS, our governing institution, decided not to sign Addgene deposit agreement. It is thus now indicated that pBL plasmids are available upon request.

Line 407: “mappy” has been replaced by “minimap2”.

Line 479: rtracklayer R package version used has been added.

Lines 600-602: the names of the softwares used for sort-seq data analysis have been added.

Methods have been updated (lines 453-464, 484-504, 521-539, 607-613) to take into account the new analyses performed in response to reviewers’ comments.

Data and code availability statements (lines 615-631 and 633-635, respectively) have been updated to comply with Nature Communications’ instructions.

Lines 803-807 and 809-810: Author contributions and Competing interests statements have been added.

Figure legends:

- “The typical timeline is indicated” has been added in Fig. 1a caption.
- “A linear regression model (coloured line)” has been added in Fig. 1c caption.
- Since the X-axis scale of Fig. 4b-f has been modified in response to Reviewer#2’s comment, the sentence “Please note that a few values standing outside the x-axis scales were omitted for clarity; the full range of values is presented for all telomeres in Supplementary Figs. 15-19” has been removed from Fig. 4b-f caption as the full range of values is now visible.

Figures have been adjusted to comply with Nature Communications formatting instructions.

Because of the addition of new Supplementary Figures, former Supplementary Figs. 1-23 are now Supplementary Figs. 3-25; Supplementary Fig. 24 is now Supplementary Fig. 27.

Response to reviewers' comments

- Reviewer #1 (Remarks to the Author):

My initial assessment of this manuscript was already positive. The authors have now addressed the concerns I had with additional analyses and experiments, in a satisfactory way. I thus support the publication of the manuscript.

Reviewer #1 (Remarks on code availability):

Although I checked the presence of the codes and the recent modifications, I do not have the expertise to review the codes.

- Reviewer #2 (Remarks to the Author):

Within a relatively short time, the authors have fully addressed all the points I raised and related points raised by Reviewer #1. I congratulate the authors for their excellent work.

- Reviewer #3 (Remarks to the Author):

The authors have included a more balanced discussion of the advantages of Nanotiming and decreased the emphasis on cost as an advantage. I am happy to support the publication of this excellent manuscript.

We thank all three reviewers for their comments.